# ISCA1 is essential for mitochondrial Fe$_4$S$_4$ biogenesis *in vivo*

Lena Kristina Beilschmidt[1,2,3,4,5], Sandrine Ollagnier de Choudens[6,7,8], Marjorie Fournier[1,2,3,4,†], Ioannis Sanakis[9], Marc-André Hograindleur[6,7,8,10], Martin Clémancey[7,8,10], Geneviève Blondin[7,8,10], Stéphane Schmucker[1,2,3,4,5], Aurélie Eisenmann[1,2,3,4,5], Amélie Weiss[1,2,3,4], Pascale Koebel[1,2,3,4], Nadia Messaddeq[1,2,3,4], Hélène Puccio[1,2,3,4,5] & Alain Martelli[1,2,3,4,5,†]

Mammalian A-type proteins, ISCA1 and ISCA2, are evolutionarily conserved proteins involved in iron–sulfur cluster (Fe–S) biogenesis. Recently, it was shown that ISCA1 and ISCA2 form a heterocomplex that is implicated in the maturation of mitochondrial Fe$_4$S$_4$ proteins. Here we report that mouse ISCA1 and ISCA2 are Fe$_2$S$_2$-containing proteins that combine all features of Fe–S carrier proteins. We use biochemical, spectroscopic and *in vivo* approaches to demonstrate that despite forming a complex, ISCA1 and ISCA2 establish discrete interactions with components of the late Fe–S machinery. Surprisingly, knockdown experiments in mouse skeletal muscle and in primary cultures of neurons suggest that ISCA1, but not ISCA2, is required for mitochondrial Fe$_4$S$_4$ proteins biogenesis. Collectively, our data suggest that cellular processes with different requirements for ISCA1, ISCA2 and ISCA1–ISCA2 complex seem to exist.

[1] IGBMC (Institut de Génétique et de Biologie Moléculaire et Cellulaire) Translational Medicine and Neurogenetics Department, 67404 Illkirch, France. [2] Inserm U596, 67404 Illkirch, France. [3] CNRS, UMR7104, 67404 Illkirch, France. [4] Université de Strasbourg, 67000 Strasbourg, France. [5] Collège de France, Chaire de Génétique Humaine, 67404 Illkirch, France. [6] CEA/DRF/BIG/CBM/BioCat, 38054 Grenoble, France. [7] CNRS UMR 5249, LCBM, 38054 Grenoble, France. [8] Université Grenoble Alpes, LCBM, 38054 Grenoble, France. [9] NCSR, Demokritos, Institut of Materials Science, Attiki, Greece. [10] CEA/DRF/BIG/CBM/pmb, 38054 Grenoble, France. † Present addresses: Sir William Dunn School of Pathology, University of Oxford, Oxford OX1 3RE, UK (M.F.); Rare Disease Research Unit, Pfizer Inc., Cambridge, Massachusetts 02139, USA (A.M.). Correspondence and requests for materials should be addressed to H.P. (email: hpuccio@igbmc.fr) or to A.M. (email: alain.martelli@pfizer.com).

I ron–sulfur clusters (Fe–S) are ancient and essential cofactors that participate in a number of cellular processes ranging from mitochondrial respiration to DNA metabolism[1,2]. In eukaryotes, *de novo* Fe–S biogenesis takes place within mitochondria and relies on proteins that are highly conserved, with homologues found in bacteria and yeast. During *de novo* Fe–S biogenesis, a cysteine desulfurase complex, consisting of NFS1 and ISD11, provides the inorganic sulfur and binds the scaffold protein ISCU on which the cluster is assembled. Frataxin (FXN), the protein deficient in Friedreich ataxia, interacts with the ternary ISCU–NFS1–ISD11 complex and controls both iron entry and sulfur transfer to ISCU by regulating the cysteine desulfurase activity[3–6]. Electrons that are required for Fe–S synthesis are most likely provided by the ferredoxin FDX2 (ref. 7). After *de novo* assembly, the cluster on ISCU is transferred to mitochondrial acceptor proteins via a late-acting Fe–S machinery involving poorly characterized accessory proteins[1,2]. In parallel, a sulfur compound, most likely derived from glutathione, is provided by the mitochondrial Fe–S machinery and is exported to the cytosol by the ABCB7 transporter where it is used as substrate to elaborate Fe–S for extra-mitochondrial acceptor proteins[8].

A-type proteins are evolutionarily conserved proteins involved in Fe–S biogenesis. In bacteria, several A-type proteins can co-exist. *Escherichia coli* encodes four A-type proteins: IscA encoded by the *isc* operon[9], SufA encoded by the *suf* operon[10], ErpA[11] and NfuA[12,13]. In yeast *Saccharomyces cerevisiae* and in mammals, two A-type proteins, ISCA1 (yeast Isa1) and ISCA2 (yeast Isa2), are nuclear-encoded and addressed to mitochondria[14,15]. Despite their presence throughout phyla, the role of A-type proteins remains unclear.

Bacterial A-type proteins were reported to bind either iron or Fe–S, and accordingly to display a role either as iron donor[16,17], in particular during Fe–S cluster assembly on IscU, or as Fe–S carrier/scaffold proteins that can provide Fe–S to acceptor proteins[10,12,13,18–20]. Furthermore, under aerobic condition in *E. coli*, only the double *iscA sufA* deletion is lethal, suggesting that bacterial A-type proteins are functionally redundant[21–23].

In *S. cerevisiae*, Isa1 and Isa2 single or double deletion strains are viable and were initially reported to present both mitochondrial and cytosolic Fe–S defects[14,24,25]. However, Isa1 and Isa2, together with their binding partner Iba57, were more recently shown to be specifically required for the maturation of mitochondrial $Fe_4S_4$ proteins[26,27]. Interestingly, only $\Delta$Isa1 yeast strain was shown to be complemented by *E. coli* A-type proteins (IscA, SufA or ErpA)[27], thus supporting the non-redundant function of eukaryotic A-type proteins. In mammals, human ISCA1 and ISCA2 knockdown experiments in HeLa cells suggest that both proteins may function together during the late biogenesis of mitochondrial $Fe_4S_4$ (ref. 15). Recent *in vitro* biochemical studies using recombinant human ISCA1 and ISCA2 demonstrated a cluster transfer and a protein–protein interaction between human glutaredoxin GLRX5 and ISCA1 or ISCA2 (ref. 28). Furthermore, a human ISCA1–ISCA2 heterodimeric complex was shown to act as a platform to assemble a $Fe_4S_4$ cluster from two $Fe_2S_2$ clusters present on GLRX5 dimer *in vitro*[29], thus providing a potential mechanism by which ISCA1 and ISCA2 could supply $Fe_4S_4$ *in vivo*.

To get further insights into the function of eukaryotic ISCA1 and ISCA2, we use the murine proteins and performed multiple complementary *in vitro* and *in vivo* characterizations. Although both proteins display key characteristics of Fe–S carriers *in vitro*, *in vivo* data provide evidence that only ISCA1 is essential for mitochondrial $Fe_4S_4$ proteins under defined physiological conditions in skeletal muscle or primary neuronal cells, while ISCA2 is dispensable. Together, our results suggest that ISCA1 and ISCA2 do not necessarily act as a complex to provide mitochondrial $Fe_4S_4$, and that a dynamic network probably exists *in vivo*.

## Results

**Mouse ISCA1 and ISCA2 are $Fe_2S_2$-containing proteins.** Mouse *Isca1* and *Isca2* were cloned to produce the respective recombinant proteins in bacteria. Whereas pure His-ISCA2 was obtained under anaerobic condition, His-ISCA1 could not be purified under this condition due to precipitation of the recombinant protein during purification, and could only be purified aerobically. On analytical Superdex-75 column, His-ISCA2 was eluted mainly as a single symmetric peak corresponding to a dimer (theoretical mass of the monomer: 13,554 Da) while His-ISCA1 behaved as a mixture of dimeric (main) and tetrameric (minor) forms (theoretical mass of the monomer: 13,898.8 Da) (Supplementary Fig. 1A). After purification both proteins are coloured. Chemical analysis revealed roughly stoichiometric amounts of iron and sulfide for both proteins with an average of 0.9 iron and 0.9 sulfur atom/monomer for His-ISCA2 and 0.9 iron and 0.8 sulfur atom/monomer for His-ISCA1, suggesting the presence of an iron–sulfur cluster. The UV-visible spectra of both proteins are in agreement with this hypothesis with absorption bands at 420 and 320 nm for both proteins and additional shoulders at 460 and 550 nm for His-ISCA2 (Supplementary Fig. 1B), suggesting the presence of a $Fe_2S_2$ cluster. The 4.2 K Mössbauer spectra of a $^{57}$Fe-labelled His-ISCA2 and $^{57}$Fe-labelled His-ISCA1 consisted of one symmetric quadrupole doublet, representing almost 100% of the iron content, whose parameters ($\delta = 0.27 (\pm 0.01)$ mm s$^{-1}$ and $\Delta E_Q = 0.53 (\pm 0.03)$ mm s$^{-1}$ for ISCA2; $\delta = 0.28(1)$ mm s$^{-1}$ and $\Delta E_Q = 0.50(2)$ mm s$^{-1}$ for His-ISCA1), unambiguously demonstrated the presence of a $[Fe_2S_2]^{2+}$ (S = 0) cluster (Fig. 1a). We further analysed both ISCA1 an ISCA2 clusters by EPR spectroscopy. After reduction with dithionite, the initial EPR-silent ISCA2 was converted to a S = 1/2 species, characterized by a rhombic EPR signal with g values at $g_1 = 1.99$, $g_2 = 1.97$ and $g_3 = 1.91$ (Fig. 1b). Temperature dependence and microwave power saturation properties of the signal were in agreement with a $[Fe_2S_2]^+$. The signal integrated for 20% of total iron. A similar EPR signal, integrating for 15% of total iron, was obtained using dithiothreitol (DTT), showing that the low yield of the Fe–S under the +1 oxidation state was not due to a strong reducing treatment that can partially destroy the cluster[18]. Similar results were obtained for His-ISCA1 (Fig. 1b). Collectively, our data indicate that both as-purified His-ISCA1 and His-ISCA2 bind an oxidized $Fe_2S_2$ under a +2 oxidation state.

To ensure that the $Fe_2S_2$ cluster on ISCAs do not result from degradation of a $Fe_4S_4$ cluster during purification steps, we performed *in cellulo* Mössbauer experiments by comparing Fe–S content of bacteria overexpressing ISCA1 or ISCA2 with control bacteria cultured under the same conditions. Despite some variations, all control spectra (Fig. 1c, upper panels) were comparable to those previously published in the literature[30–34]. The main difference between the spectrum recorded on bacteria overexpressing ISCA2 proteins (Fig. 1c, lower right panel) and that of the control (Fig. 1c, upper right panel) is the additional line clearly observed at about 0.5 mm s$^{-1}$ (Fig. 1c, dashed line), suggesting the contribution of $[Fe_2S_2]^{2+}$ clusters. Satisfying simulations were obtained for both spectra (see Methods Section, Fig. 1c, Table 1). Twenty per cent of the total iron content were observed as $[Fe_2S_2]^{2+}$ clusters for ISCA2 samples. Similar results were obtained on four different sample preparations for ISCA2. Furthermore, using a more efficient expression system (see Methods section), we were able to increase the spectral

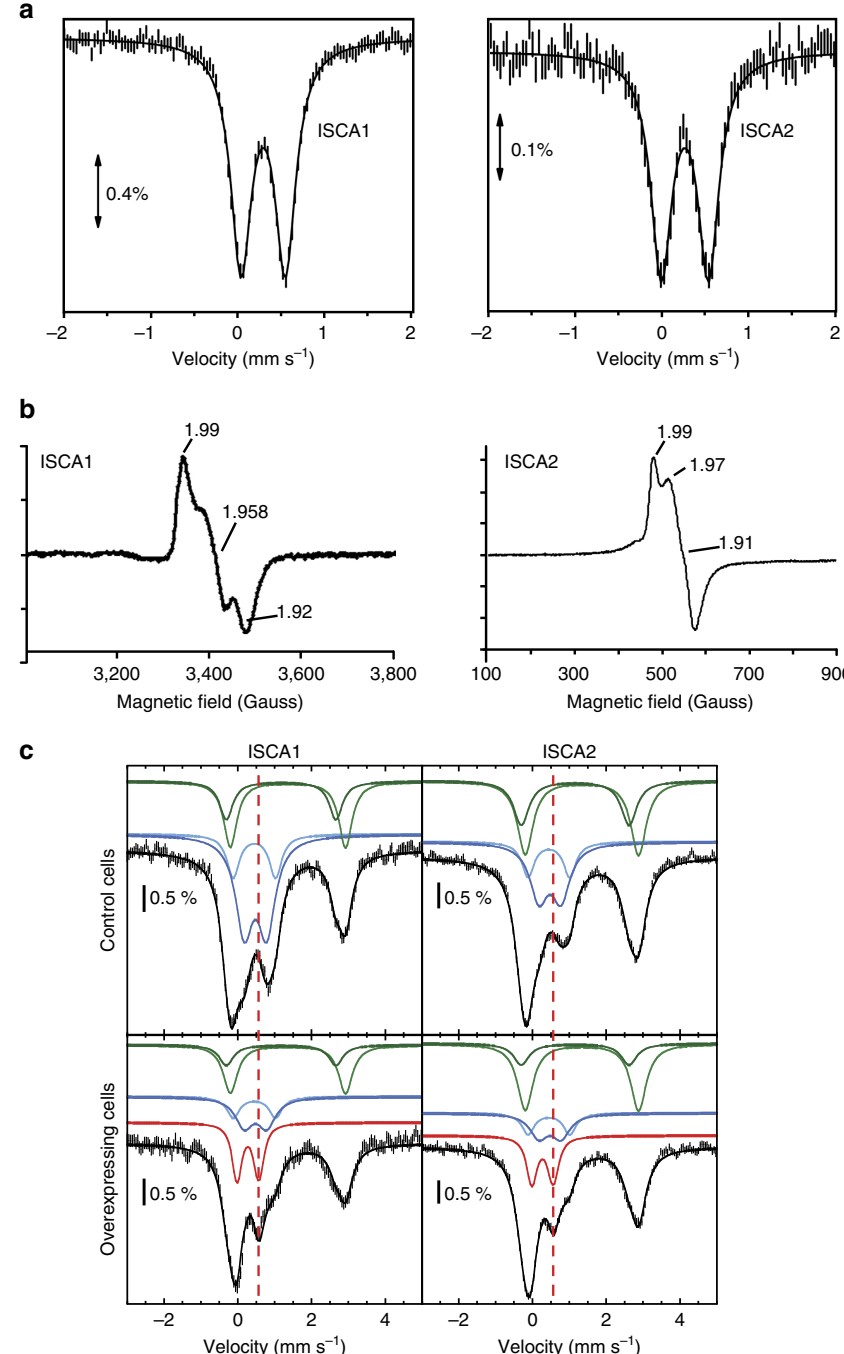

**Figure 1 | Spectroscopic characterization of ISCA1 and ISCA2.** (**a**) Mössbauer spectra of the purified $^{57}$Fe/S-ISCA1 (1.1 mM, 1 Fe and 0.9 S/protein) and $^{57}$Fe/S-ISCA2 (380 μM, 0.7 Fe and 0.65 S/protein). Mössbauer spectra were recorded at 4.2 K in a magnetic field of 83 and 60 mT applied perpendicular to the γ-beam for ISCA1 and ISCA2, respectively. The solid lines correspond to simulation of the experimental spectra. (**b**) X-band EPR spectra of as-isolated ISCA1 (250 μM) and ISCA2 (500 μM) after reduction with 1 mM dithionite for 10 min. (**c**) Mössbauer spectra recorded on whole control cells (upper panels) and ISCA1 or ISCA2-overexpressing cells (lower panels) at 5.5 K using a 60 mT external magnetic field applied parallel to the γ-beam. Experimental spectra are shown with hatched marks and simulations are overlaid as solid black lines. Five components were used for simulation: HS Fe$^{II}$ (light and dark green), Fe$_4$S$_4$ clusters and LS ferrous haems (light blue), Fe$^{III}$ NP (dark blue) and [Fe$_2$S$_2$]$^{2+}$ (red) (see Table 1 for parameter values).

intensity signal, and 30% of the total iron content was observed as [Fe$_2$S$_2$]$^{2+}$ clusters for ISCA2 (Supplementary Fig. 1C and Supplementary Table 1). Comparable results were obtained by overexpressing ISCA1. However, due to the low solubility of the ISCA1 protein in these conditions, the signal was more variable and a significant signal for [Fe$_2$S$_2$]$^{2+}$ clusters could only be observed in three performed experiments with the best one

showing up to 24% contribution of the [Fe$_2$S$_2$]$^{2+}$ clusters (Fig. 1c, lower left panel, Table 1). For both induced ISCA cell samples, the amount of additional Fe–S cluster is larger than the experimental uncertainty and comparable to that measured in other studies[30,31]. A perusal of Table 1 reveals that the total contribution of high-spin (HS) Fe$^{II}$ and Fe$_4$S$_4$ components did not vary significantly between control and induced ISCA cell

**Table 1 | Values of the nuclear parameters and contributions for the five components issued from simulations of the spectra\*.**

| Species | $\delta$(mm s$^{-1}$) | $\Delta E_Q$(mm s$^{-1}$) | $\Gamma$(mm s$^{-1}$)† | % in control cell samples | | % in induced ISCA cells | |
|---|---|---|---|---|---|---|---|
| *ISCA1* | | | | | | | |
| HS Fe$^{II}$ | 1.36 | 3.13 | 0.43 | 25 | *39* | 30 | *43* |
| HS Fe$^{II}$ | 1.18 | 2.97 | 0.43 | 14 | | 13 | |
| [Fe$_4$S$_4$]$^{2+}$ and LS Fe$^{II}$ haem | 0.45 | 1.15 | 0.40 | 15 | *61* | 12 | *34* |
| Fe$^{III}$ NP | 0.48 | 0.61 | 0.57 | 45 | | 22 | |
| [Fe$_2$S$_2$]$^{2+}$‡ | 0.27 | 0.50 | 0.28 | <2 | *<4* | 12 | *24* |
| | 0.28 | 0.68 | 0.28 | <2 | | 12 | |
| | | | | | | | |
| *ISCA2* | | | | | | | |
| HS Fe$^{II}$ | 1.36 | 3.13 | 0.52 | 36 | *57* | 41 | *54* |
| HS Fe$^{II}$ | 1.18 | 2.97 | 0.52 | 21 | | 13 | |
| [Fe$_4$S$_4$]$^{2+}$ and LS Fe$^{II}$ haem | 0.45 | 1.15 | 0.41 | 13 | *43* | 10 | *26* |
| Fe$^{III}$ NP | 0.48 | 0.61 | 0.60 | 30 | | 16 | |
| [Fe$_2$S$_2$]$^{2+}$‡ | 0.27 | 0.50 | 0.32 | <2 | *<4* | 10 | *20* |
| | 0.28 | 0.68 | 0.32 | <2 | | 10 | |

\*Shown in Fig. 1c Italic numbers indicate the overall contributions of HS Fe$^{II}$ species, of the two components with 0.4–0.5 mm s$^{-1}$ isomer shift values (Fe$_4$S$_4$ clusters, LS ferrous haems and Fe$^{III}$ NP), and of the two sites of the Fe$_2$S$_2$ clusters. Uncertainties are $\pm$0.02 mm s$^{-1}$ on $\delta$, $\Delta E_Q$ and $\Gamma$ and $\pm$3 on the percentage of each site.
†The linewidth for HS Fe$^{II}$ components were constrained to be equal. The same holds for the two sites of the [Fe$_2$S$_2$]$^{2+}$ clusters.
‡Isomer shift and quadrupole splitting values are those published for the [Fe$_2$S$_2$]$^{2+}$ cluster of $^{Nif}$IscA (ref. 53) and were fixed. The two sites were assumed to contribute in a 1:1 ratio.

samples. Conversely, the contributions of the component corresponding to Fe$^{III}$ nanoparticles (Fe$^{III}$ NP) exhibited an important decrease (23% for ISCA1 and 14% for ISCA2), suggesting that the oxidized Fe$_2$S$_2$ detected in the ISCA-overexpressing cells mainly originated from this pool of iron. Since oxygen-sensitive Fe$_4$S$_4$ clusters of proteins were shown to be detectable by *in cellulo* Mössbauer spectroscopy using bacteria overexpressing the corresponding protein[31,35–37], our results indicate that mouse recombinant ISCA1 and ISCA2 are produced as Fe$_2$S$_2$ proteins in bacteria.

**Mouse ISCA1 and ISCA2 behave as Fe–S carriers *in vitro*.** An Fe–S carrier protein is defined by its capacity (i) to accept Fe–S from a scaffold protein and (ii) to transfer its Fe–S to target proteins. The Fe–S transfer between the ISCU scaffold and ISCAs was first investigated using the His-tagged mouse ISCA and ISCU proteins. Fe$_2$S$_2$-ISCA1 or Fe$_2$S$_2$-ISCA2 was incubated, under anaerobic condition, with one equivalent of apo-ISCU. After desalting and separation, ISCA2 (initially with 0.6 Fe and 0.6 S per monomer) was still containing 0.4 Fe and 0.45 S/monomer, and the UV-visible spectrum (absorption bands at 320, 420 and 460 nm) was very similar to the one before incubation, showing that no cluster transfer occurred (Fig. 2a, upper left panel). Accordingly, no iron or sulfur could be detected on ISCU and the UV-visible spectra of ISCU were similar before and after incubation (Fig. 2a, lower left panel). A similar result was obtained using Fe$_2$S$_2$-ISCA1 as Fe–S donor (Supplementary Fig. 1D). However, when holo-ISCU was incubated with apo-ISCA2, cluster transfer was observed, with changes in the UV-visible spectra (Fig. 2a, right panels), a loss of 98% of the iron on ISCU, and 0.7 Fe/monomer and 0.6 S/monomer on ISCA2 after separation. Owing to the instability of apo-ISCA1, the transfer between Fe$_2$S$_2$-ISCU and apo-ISCA1 could not be conclusively assessed.

The capacity of ISCA proteins to transfer their cluster to Fe$_2$S$_2$ and Fe$_4$S$_4$ target proteins was then investigated using *E. coli* apo ferredoxin (Fdx) and mouse mitochondrial aconitase (ACO2), respectively. After 30 min incubation of apo-Fdx with an excess of Fe$_2$S$_2$-ISCA2, the characteristic UV-visible spectrum of Fe$_2$S$_2$-Fdx, with $\lambda_{max}$ at 415 and 460 nm, was obtained (Fig. 2b). Reduction of the ISCA2-Fdx mixture with dithionite and further analysis by EPR unambiguously demonstrated the formation of

holo-Fdx, since a $S = 1/2$ EPR signal characteristic of the reduced [Fe$_2$S$_2$]$^+$ cluster of Fdx was observed (Supplementary Fig. 1E)[38]. Similar results were obtained when Fe$_2$S$_2$-ISCA1 was used as a Fe–S donor protein (Supplementary Fig. 1F). Apo-ACO2 was incubated anaerobically with a 3.5-fold molar excess of either native ISCA1 or ISCA2, and transfer was monitored by measuring ACO2 enzyme activity. Upon incubation with Fe$_2$S$_2$-ISCA1 or Fe$_2$S$_2$-ISCA2, ACO2 activity was progressively increasing and reached more than 70% and 80% of the activity of a chemically reconstituted ACO2 after 30 min incubation, respectively (Fig. 2c). Fe–S transfer was observed only in the presence of a reducing agent such as DTT in agreement with the requirement of electrons to generate Fe$_4$S$_4$ from Fe$_2$S$_2$ clusters[18]. These data clearly show that native ISCA proteins can transfer their Fe–S to both Fe$_2$S$_2$ and Fe$_4$S$_4$ apo-proteins.

Together, our results show that both ISCA1 and ISCA2 have key characteristics of Fe–S carrier proteins *in vitro*.

**ISCA1 and ISCA2 have distinct interacting partners.** To further define the process in which ISCA proteins are involved, we sought interacting partners by performing immunoprecipitation (IP) experiments coupled to multidimensional protein identification technology (MudPIT). IP were carried out using HeLa mitochondrial extracts expressing mouse ISCA1, ISCA2, IBA57, FDX2 or GLRX5 with a C-terminal Flag epitope, respectively. MudPIT analysis was then performed on crude IP elutions. Specific interacting partners were filtered with high stringency criteria, to be enriched ten times minimum over both control-IP and mitochondrial-enriched extracts. Their relative abundance was estimated by label-free quantification analysis, based on normalized spectral abundance factor (NSAF). The full data set can be found in Supplementary data file. To get insights on the role of ISCA proteins in Fe–S biogenesis, we restricted our current analysis to the identification of proteins known to be involved in the Fe–S pathway (Fig. 3a, Supplementary Data 1). As previously described[27,29], a reciprocal interaction between ISCA1 and ISCA2 was found (Fig. 3a). The interaction was confirmed in an independent ISCA1-Flag IP in N2a cells, followed by western blot analysis with an ISCA2 antibody (Fig. 3b). Interestingly, ISCA2 and IBA57 reciprocally co-purified, whereas no interaction was detected between IBA57 and ISCA1 (Fig. 3a). The specific interaction between ISCA2 and IBA57

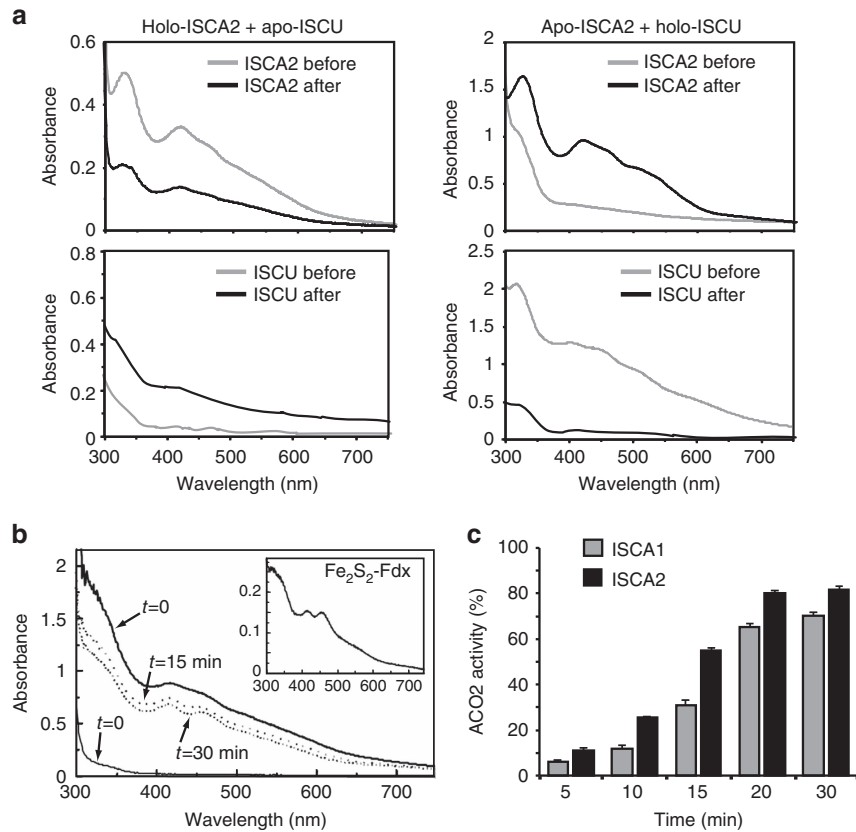

**Figure 2 | Fe–S transfer experiments using ISCA1 and ISCA2 recombinant proteins.** (**a**) UV-visible spectra obtained by mixing $Fe_2S_2$-ISCA2 with apo-ISCU (left panels) and apo-ISCA2 and $Fe_2S_2$-ISCU (right panels). (**b**) Monitoring of the apo-ferredoxin (thin line, 100 μM) and as-isolated $Fe_2S_2$-ISCA2 (bold line, 200 μM) mixture after 15 min (dotted line) and 30 min (dashed line) incubation. The inset shows the UV-visible spectrum of the $Fe_2S_2$-ferredoxin (20 μM) obtained after separation onto a NiNTA column. (**c**) ACO2 (0.2 nmol) activity after incubation with tenfold excess of as-isolated ISCA1 (0.9 Fe/monomer) or ISCA2 (0.6 Fe/monomer) at 5, 10, 15, 20 and 30 min. Reconstituted ACO2 (0.2 nmol, 3.8 iron/protein) was used as positive control to set the 100% activity. Data are represented as the mean of three measurements ± s.d.

could be confirmed in IP experiments performed with N2a cells (Fig. 3b). Similarly, GLRX5 and ISCA2 reciprocally co-purified, but with rather low NSAF values in the MudPIT analysis (Fig. 3a). In agreement, a low GLRX5 signal was detected in ISCA2-Flag IP experiments with N2a cells (Fig. 3b). Finally, whereas a faint interaction (low NSAF) with FDX2 was observed in the ISCA2-Flag IP (Fig. 3a), no peptide was detected in the reverse IP, suggesting that FDX2 may rather be an indirect or weak binding partner of ISCA2. In contrast, the *de novo* Fe–S biosynthesis complex components, NFS1, ISCU and ISD11 were found in high abundance in the FDX2-Flag IP (Fig. 3a), strongly supporting the current model of FDX2 as an electron donor for *de novo* Fe–S assembly[7]. In addition, NFU1 was detected in the ISCA1-Flag IP by MudPIT (Fig. 3a), and was confirmed in the ISCA1-Flag independent IP using N2a cells (Fig. 3b). The interaction appeared specific to ISCA1 since NFU1 was neither co-immunoprecipitated with ISCA2-Flag nor IBA57-Flag (Fig. 3a,b).

Collectively, these results demonstrate a direct interaction between ISCA1 and ISCA2, as well as between ISCA2, IBA57 and GLRX5, whereas NFU1 is a partner of ISCA1 (Fig. 3c). Our results show that despite the existence of a prominent ISCA1–ISCA2 complex, distinct complexes containing either ISCA1 or ISCA2 may exist *in vivo*.

**ISCA1 is required for the maturation of mitochondrial $Fe_4S_4$.** To determine the *in vivo* functions of ISCA1 and ISCA2, we

carried out knockdown experiments by injecting recombinant adeno-associated virus (AAVs) encoding for specific shRNAs into the tibialis anterior (TA) of 4-week-old mice. A scrambled shRNA (CTL) and a shRNA directed against the expression of the ISCU scaffold protein were used as controls. Strong knockdowns were confirmed 3 weeks post injection (w.p.i.) by qRT–PCR for ISCA1 and by western blot for ISCA2 and ISCU (Fig. 4a and Supplementary Fig. 2A). At dissection, no gross muscle defect due to the viral infection was observed and histological analysis using haematoxylin-eosin staining confirmed an overall preserved muscle organization (Fig. 4b, left panels). By electron microscopy, muscle fibres were completely normal, and in particular, no mitochondrial disorganization was observed (Supplementary Fig. 2B). We next analysed Fe–S-dependent proteins in the TA muscle. The activity of succinate dehydrogenase (SDH), containing both $Fe_2S_2$ and $Fe_4S_4$ clusters, was strongly decreased in skeletal muscle after *Isca1* and *Iscu* knockdowns (Fig. 4b, right panels and Supplementary Fig. 2C). A substantial decrease of mitochondrial $Fe_4S_4$ containing proteins, including aconitase (ACO2), complex II (SDH B) and two subunits of complex I (NDUFS3 and NDUFS5) was observed by western blot after *Isca1* and *Iscu* knockdowns (Fig. 4a), in line with previously reported instability of the respective apo-proteins or complexes in the absence of Fe–S in eukaryotic cells[15,39]. In addition, a decrease in lipoic acid (LA) bound to pyruvate dehydrogenase (PDH) and α-ketoglutarate dehydrogenase (KGDH) complexes was observed after *Isca1* and *Iscu* knockdowns (Fig. 4a). Since the levels of the PDH-E2 subunit, on which LA is bound, were unchanged

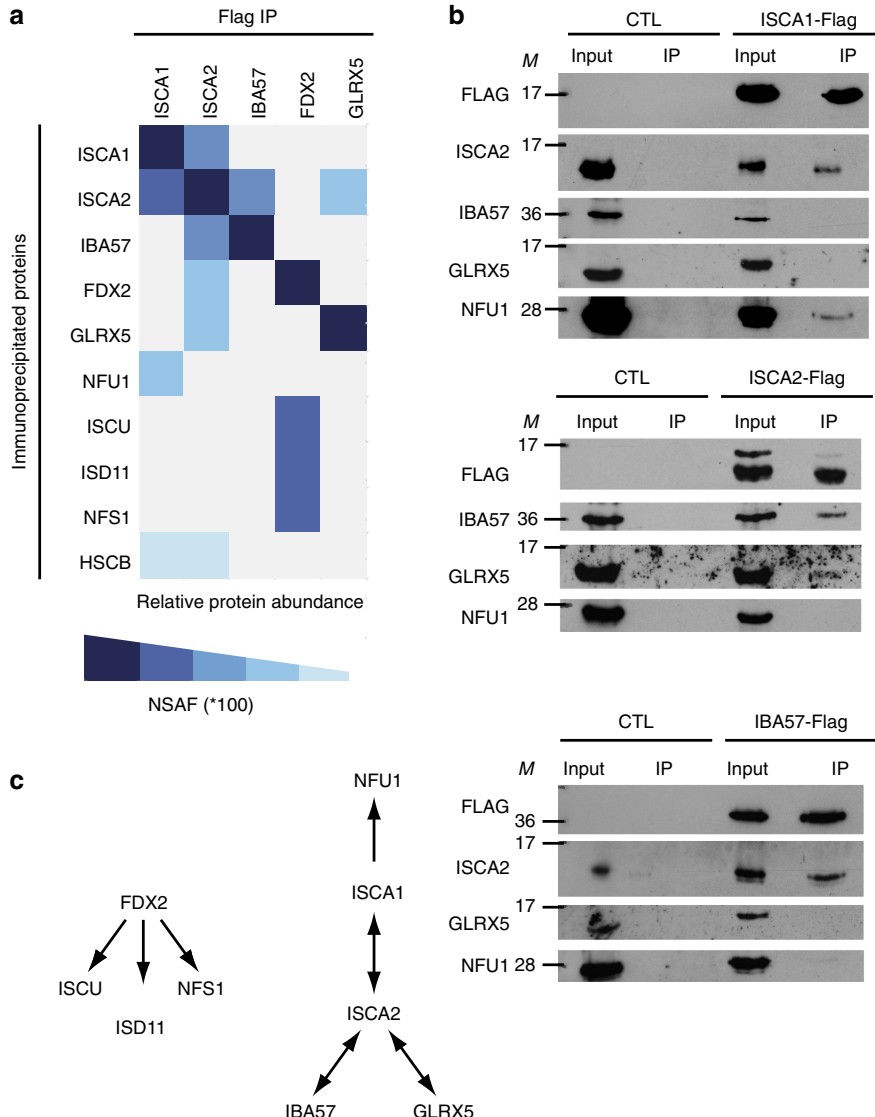

**Figure 3 | ISCA1 and ISCA2 interaction network.** (**a**) Fe–S machinery-related proteins and their relative abundance (NSAF) determined by MudPIT analysis after immunoprecipitation of the indicated Flag-tagged baits. Data represent proteins enriched 10 × over controls. The colour intensity reflects the NSAF values multiplied by 100. (**b**) Representative western blot analysis using different available antibodies (as indicated) of ISCA1-Flag, ISCA2-Flag and IBA57-Flag IPs versus control IP (CTL) obtained after transfection of N2a cells with the respective constructs ($n \geq 3$). M indicates size markers. (**c**) Interaction network within the Fe–S machinery based on western blot and MudPIT analysis.

(Fig. 4a), the observed decrease in LA reflects a defect in the $Fe_4S_4$ enzyme lipoic acid synthase (LIAS). Conversely, no decrease in the tested mitochondrial $Fe_2S_2$ enzymes (FECH and RIESKE) or in the non-Fe–S containing enzyme COX IV was observed for the *Isca1* knockdown. A decrease of $Fe_2S_2$ enzymes, in addition to $Fe_4S_4$ enzymes, was however seen with the depletion of ISCU in accordance with the central role of the scaffold protein for all Fe–S (Fig. 4a). ISCA1 depletion also had no effect on extra-mitochondrial Fe–S protein levels (ABCE1 and GPAT) (Fig. 4a). A similar pattern for the ISCA1 knockdown was observed at 6 w.p.i. (Supplementary Fig. 2D).

The specificity of the observed effects was further controlled through the expression of a shRNA-resistant version of ISCA1 (ISCA1[R]), that is, containing a silent mutation in the region targeted by the shRNA. The combination of ISCA1 knockdown with overexpression of the ISCA1[R] was validated by qRT–PCR using specific primer pairs (Supplementary Fig. 2F). All alterations in maturation of mitochondrial $Fe_4S_4$ enzymes were fully prevented by the expression of ISCA1[R] (Fig. 4c, left panel),

demonstrating the specific implication of ISCA1 in mitochondrial $Fe_4S_4$ protein maturation.

Quite surprisingly, knockdown of ISCA2 had no effect on any of the tested Fe–S proteins in muscle 3 w.p.i. (Fig. 4a) and 6 w.p.i. (Supplementary Fig. 2D). Accordingly, the SDH activity was normal on muscle sections (Fig. 4b, right panels and Supplementary Fig. 2C). Furthermore, IBA57 knockdown also did not impair any of the tested Fe–S proteins (Supplementary Fig. 2E).

To fully evaluate the potential functional redundancy between ISCA1 and ISCA2, we combined knockdown of endogenous *Isca1* with overexpression of ISCA2. Knockdown of *Isca1* and over-expression of ISCA2 were validated by qRT–PCR and western blot, respectively (Fig. 4c, right panel and Supplementary Fig. 2F). In contrast to the full prevention that was obtained with ISCA1[R] expression (Fig. 4c, left panel), ISCA2 overexpression had no effect on the maturation of mitochondrial $Fe_4S_4$ proteins after ISCA1 knockdown (Fig. 4c, right panel), hence demonstrating that ISCA2 cannot compensate for the loss of ISCA1 function

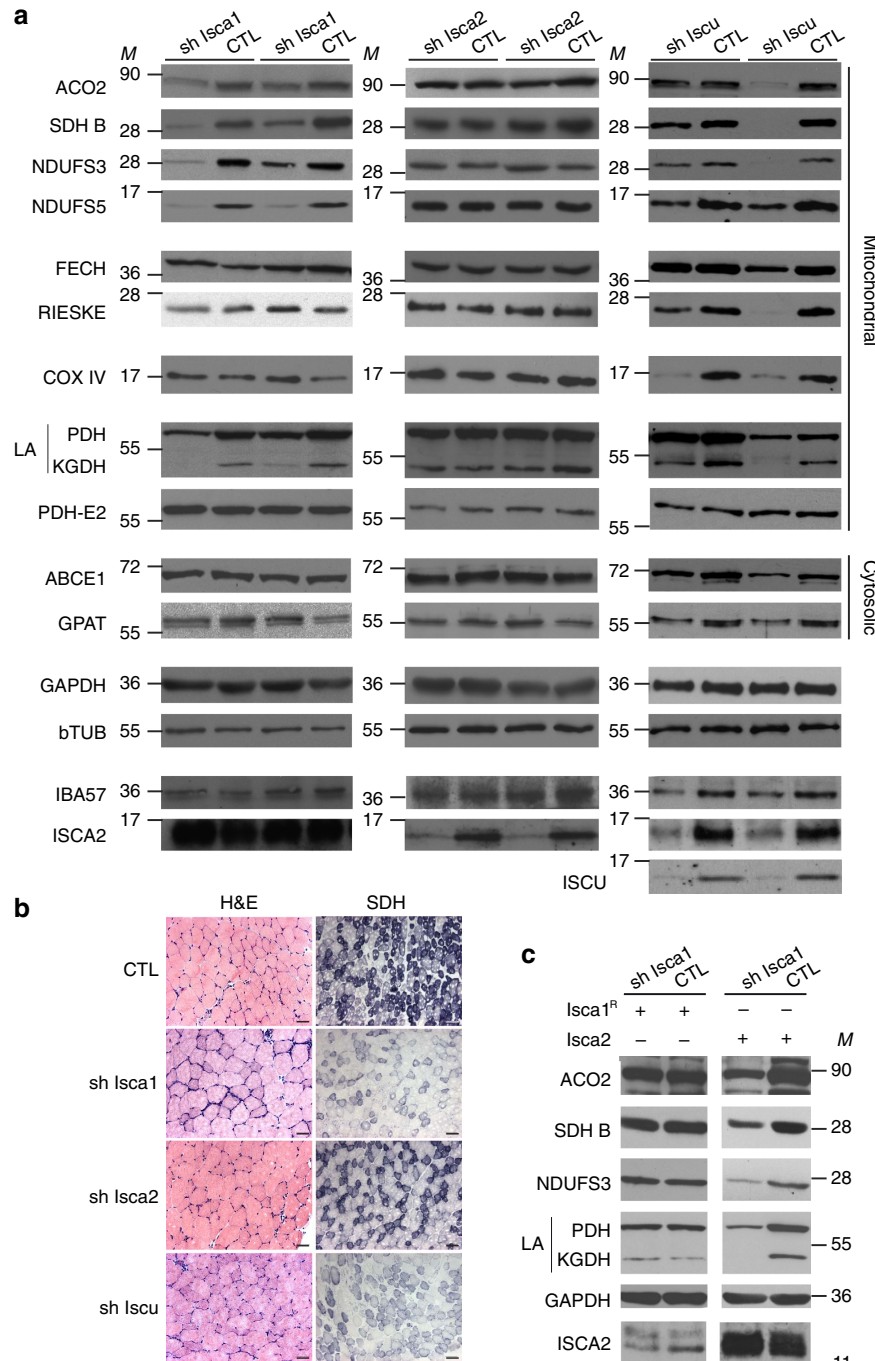

**Figure 4 | ISCA1 and ISCA2 knockdowns in skeletal muscle.** (**a**) Representative western blots of two independent samples for the indicated proteins using extracts from TA muscle at 3 w.p.i., injected with rAAV-scrambled shRNA (CTL), rAAV-shIsca1, rAAV-shIsca2 or rAAV-shIscu ($n \geq 5$). GAPDH and beta-tubuline (bTUB) were used as loading controls. M indicates size markers. (**b**) Histological analysis after haematoxylin-eosin (H&E) or SDH activity staining on cryosections from TA muscle at 3 w.p.i., injected with rAAV-scrambled shRNA (CTL), rAAV-shIsca1, rAAV-shIsca2 or rAAV-shIscu ($n = 5$). Scale bar, 50 µM (H&E) or 100 µM (SDH). (**c**) Representative western blots for the indicated proteins using extracts from TA muscle at 3 w.p.i., co-injected with rAAV-shISCA1 or rAAV-scramble shRNA (CTL) and rAAV-ISCA1[R] or rAAV-ISCA2 ($n = 4$). GAPDH was used as loading control. M indicates size markers.

*in vivo* and that ISCA1 and ISCA2 are not functionally redundant.

Together, these results provide clear evidence that ISCA1 is required for the maturation of mitochondrial $Fe_4S_4$ proteins in mature skeletal muscle, whereas ISCA2 is not essential in this tissue, under standard physiological conditions.

To evaluate the cellular function of ISCA1 and ISCA2 in another cell type, we performed knockdown experiments on primary cultures of mouse sensory neurons. Primary cultures

were efficiently transduced using AAV2/9 recombinant vectors coding for the shRNAs against *Isca1* and *Isca2*, as indicated by the GFP signal observed in transduced cells (Fig. 5a). Fifteen days post-infection, a strong acidification of the medium (yellow colour) was observed in the culture infected with *Isca1* shRNA (Fig. 5b), suggesting that ISCA1-depleted neurons are highly glycolytic and produce high levels of lactic acid due to mitochondrial dysfunction. Accordingly, the evaluation of LA levels by western blot showed a decrease of LA on α-ketogluterate

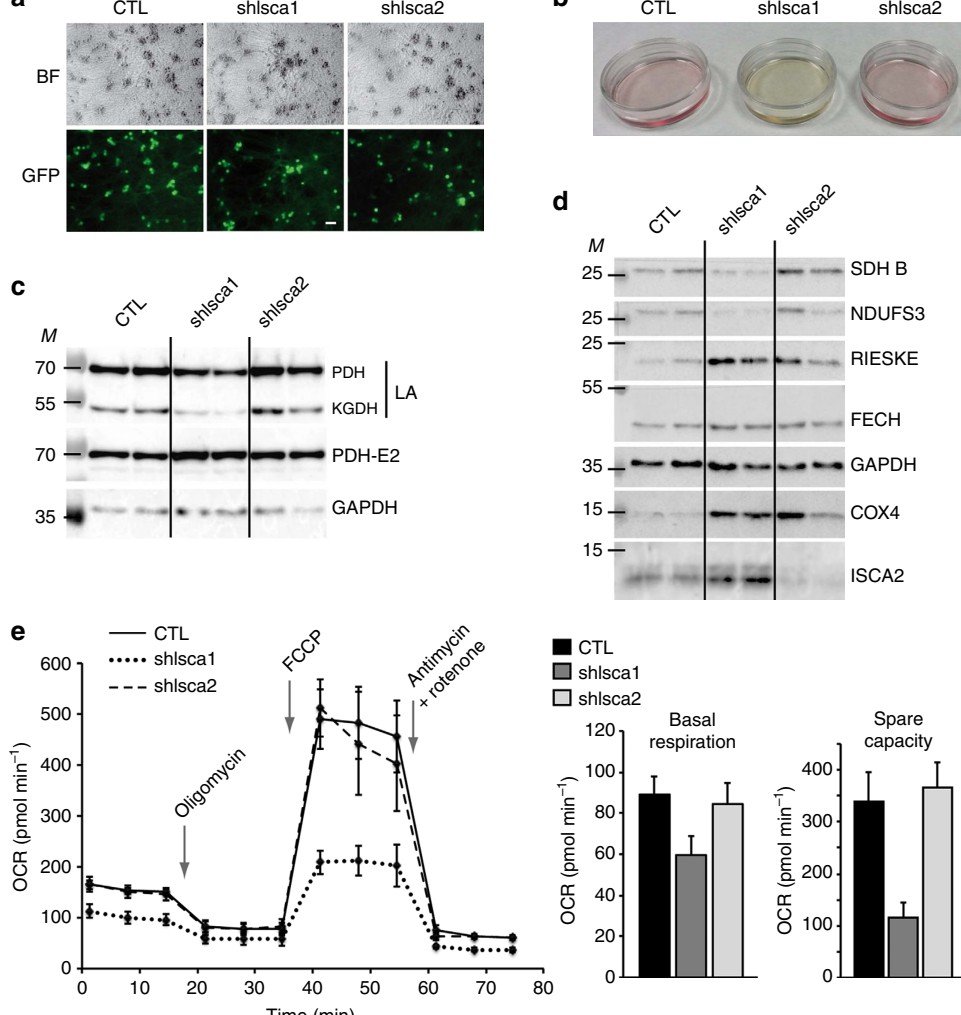

**Figure 5 | ISCA1 and ISCA2 knockdowns in primary sensory neurons.** (**a**) Brightfield and immunofluorescence (GFP) analysis of neuronal cultures 4 days after infection with rAAV-scrambled shRNA (CTL), rAAV-shIsca1 or rAAV-shIsca2. Scale bar, 100 μm. (**b**) Representative picture of the neuronal cultures 15 days after infection with rAAV-scrambled shRNA (CTL), rAAV-shIsca1 or rAAV-shIsca2 ($n = 6$). (**c**) Representative western blots of two independent samples for lipoic acid (LA) and the E2 subunit of PDH (PDH-E2) using neuronal extracts 15 days after infection with rAAV-scrambled shRNA (CTL), rAAV-shIsca1 or rAAV-shIsca2. GAPDH was used as loading control ($n = 6$). M indicates size markers. (**d**) Representative western blots of two independent samples for the indicated proteins using neuronal extracts 15 days after infection with rAAV-scrambled shRNA (CTL), rAAV-shIsca1 or rAAV-shIsca2. GAPDH was used as loading control ($n = 6$). M indicates size markers. (**e**) Representative Seahorse analysis using XF Cell Mito Stress Test Kit obtained with neurons 8 days after infection with rAAV-scrambled shRNA (CTL), rAAV-shIsca1 or rAAV-shIsca2 ($n \geq 5$). Basal respirations and spare capacities are represented as the mean ± s.d. *** $p < 0.001$. OCR: oxygen consumption rate.

dehydrogenase and PDH complexes in *Isca1* knockdowns, whereas no effect was observed in control or ISCA2-deficient neurons (Fig. 5c). Similarly, the levels of mitochondrial $Fe_4S_4$-containing proteins (SDHB and NDUFS3) were specifically reduced in *Isca1* knockdowns (Fig. 5d).

The impact of both ISCA1 and ISCA2 depletions on mitochondrial respiration was investigated on primary neurons in culture using Seahorse and the XF Cell Mito Stress Test Kit. Four days after infection with AAVs, most cells expressed GFP (Supplementary Fig. 3A). Seahorse measurement was performed 8 days post-infection. A clear and drastic reduction of both the basal respiration and the spare respiratory capacity was observed in ISCA1-deficient neurons (Fig. 5e), whereas no significant difference was detected between control and ISCA2-depleted neurons despite a significant decrease of ISCA2 protein level (Fig. 5e and Supplementary Fig. 3B). Hence, these results further demonstrate the specific role of ISCA1 in the maturation of

mitochondrial $Fe_4S_4$ that are essential for mitochondrial respiration and function.

## Discussion

Among components of the Fe–S protein assembly, the A-type ISC assembly factors are not fully characterized. In the present manuscript, we further validate the central role of ISCA1 for the maturation of mitochondrial $Fe_4S_4$ proteins while demonstrating that ISCA2 and IBA57 are not required under standard physiological conditions in two post-mitotic cell types.

Yeast Isa1 was the first eukaryotic A-type protein to be purified, however obtained as an apo-protein that could bind an $Fe_2S_2$ only after chemical reconstitution[40]. More recently, Bianci *et al.* isolated anaerobically human ISCA2 in an $Fe_2S_2$ form, but again ISCA1 was shown to contain an $Fe_2S_2$ only after chemical reconstitution[28]. Our results demonstrate for the first time that

both ISCA1 and ISCA2, as isolated (without any chemical treatment), are $Fe_2S_2$ proteins, and that the $Fe_2S_2$ clusters do not result from $Fe_4S_4$ degradation during purification based on our *in cellulo* experiments. Furthermore, our results are in agreement with the observation that ISCA proteins can function as Fe–S carriers. Accordingly, *in vivo*, we showed specific interactions of ISCA1 and ISCA2 with proteins of the late mitochondrial machinery, namely GLRX5, IBA57 and NFU1 proteins. Although a strong interaction of ISCA1 with ISCA2 was observed, consistent with previously reported data obtained in yeast or using the recombinant human proteins[27,29], the functional relevance of this interaction in cells remains to be determined. Interestingly, the distinct respective interaction between ISCA2 and IBA57 contrasts with results published in yeast where both Isa1 and Isa2 interact with Iba57 protein by co-IP[26]. However, this specific ISCA2–IBA57 interaction correlates with our *in vivo* data since neither ISCA2 nor IBA57 knockdown led to a phenotype associated with Fe–S metabolism. These results point to the complexity of the late Fe–S cluster assembly machinery, and that dynamic networks probably exist. Of note NFU1 was reported to bind $Fe_4S_4$ *in vitro*[41], and recently identified mutation in NFU1 lead to multiple mitochondrial dysfunction syndrome[42,43] with respiratory complex and LA deficiencies that are relying on mitochondrial $Fe_4S_4$ biogenesis. Through the distinct interaction of ISCA1 with NFU1, we can therefore speculate that NFU1 might be involved with ISCA1 in $Fe_4S_4$ assembly and/or delivery to mitochondrial proteins.

Finally, the different phenotypes highlighted between ISCA1 and ISCA2 under our conditions contrast with data obtained in HeLa cells in which similar phenotypes were observed for ISCA1 and ISCA2 knockdowns[15]. However, the use of multiple transfections with specific siRNAs in HeLa cells that were required to observe a phenotype may have hampered the identification of primary specific effects. The role of ISCA2 *in vivo* remains difficult to define as no phenotype was revealed under our conditions. The fact that a phenotype was found using HeLa cells, which are highly proliferative cells, may suggest that ISCA2 is required in Fe–S biogenesis during cell division, which may be in agreement with its implication during development[44–46] or depending on the physiological state of the cell (that is, proliferative, post-mitotic, oxidative stress, hypoxia, and so on). Interestingly, recently identified mutations in ISCA2 and IBA57 in human lead to severe genetic disorders in young children that may reflect a specific implication of these two proteins during development[44–46]. Differences between ISCA1 and ISCA2 are also highlighted by the fact that holo-ISCA proteins were obtained under different conditions: ISCA2 under anaerobiosis, whereas ISCA1 should be purified under aerobiosis. This may suggest that ISCA clusters have a different response to redox conditions. Whether such properties are related to specific physiological functions for each protein remains to be determined. Collectively, our results with the results in the literature strongly suggest that late mitochondrial Fe–S biogenesis is a complex and dynamic system that may have tissue and temporal specificity.

## Methods

**Cloning and plasmid constructs.** Full length *Isca1*, *Isca2*, *Glrx5*, *Fdx2* and *Iba57* (*C1orf69* homolog) cDNAs were PCR amplified from a bank of mouse heart cDNAs using specific primers and cloned in pCDNA3.1 + (zeo) for the mammalian expression of the Flag-tagged proteins. Mature forms of *Isca1* and *Isca2* were cloned into pET-11a (Novagen) for the bacterial expression of the N-terminal His-tagged recombinant proteins or into pACYC-DUET-1 (Novagen) for the bacterial expression of non-tagged recombinant proteins. Specific shRNAs against mouse *Isca1*, *Isca2*, *Iba57* or *Iscu* and scrambled shRNAs were cloned into a modified pENTR1A (Invitrogen Life Technologies) vector containing the U6 promoter. ShRNAs were subcloned into pAAV vector containing a CMV-eGFP cassette and

AAV2-specific ITR sequences using Gateway technology. Mouse ISCA1 encoded by a cDNA resistant to the shRNA ($ISCA1^R$) was obtained by directed mutagenesis and cloned into pAAV-CMV (Stratagene). Site-directed mutagenesis was carried out using specific primers and the Phusion high-fidelity DNA polymerase (Finnzymes) with the GC-Buffer and following the manufacturer's recommendations. All constructs were verified by Sanger sequencing. Oligonucleotide sequences are listed in Supplementary Table 2.

**Production and purification of recombinant proteins.** ISCA1 and ISCA2 were obtained by growing *E. coli* BL21(DE3)/pET-ISCA1 or pET-ISCA2 in LB medium containing 100 µg ml$^{-1}$ ampicillin, at 37 °C. ISCA2 expression was induced for 4 h by adding 0.5 mM isopropyl β-D-thiogalactoside (IPTG) to an exponentially growing culture. ISCA1 protein induction was performed at 20 °C overnight. Bacterial pellets were resuspended in buffer A (50 mM Tris-HCl pH 8, 50 mM NaCl, 0.1% TritonX100, 5 mM DTT, 1 mM PMSF) and sonicated before ultra-centrifugation (90 min, 144,000*g*, 4 °C). Supernatants were treated with DNAse I and 10 mM $MgCl_2$ at 4 °C for 45 min, and after centrifugation (30 min, 13,000*g*, 4 °C), soluble proteins were loaded onto a 5 ml Ni-NTA affinity column (Qiagen) equilibrated with buffer B (50 mM Tris-HCl pH 8, 50 mM NaCl, 0.1% TritonX100). Fractions containing ISCA proteins were pooled after elution with buffer B containing 0.2 M imidazole. This protocol allowed the purification of approximately 5 and 40 mg of ISCA1 and ISCA2, respectively. For Mössbauer experiments, bacteria expressing ISCA1 or ISCA2 were grown aerobically in minimal M9 $^{57}Fe$-enriched medium (35 µM) supplemented with 2 mM $MgSO_4$, 0.4% glucose, 2 µg ml$^{-1}$ thiamine, 1 mM $CaCl_2$. $^{57}Fe$ (Cambridge Isotopes) was prepared by resuspending $^{57}Fe$ powder in concentrated HCl:$HNO_3$ (1:1 ratio) in order to get a 500 mM solution. Proteins expressions as well as purifications were performed as described above.

Mouse aconitase (ACO2) was purified from an *E. coli* BL21(DE3) strain containing pRARE2 plasmid (encoding rare tRNAs) and pGEX-ACO2 that were grown in LB at 37 °C for 3 h after addition of 0.5 mM IPTG. The bacterial pellet was resuspended in buffer C (PBS supplemented with 0.1% TritonX100 and 1 mM PMSF) and sonicated before ultracentrifugation (90 min, 144,000*g*, 4 °C). The supernatant was treated with DNAse I and 10 mM $MgCl_2$ at 4 °C for 45 min, and after centrifugation (30 min, 13,000*g*, 4 °C), soluble proteins were loaded onto a 5 ml GST-trap affinity column (Qiagen) equilibrated with buffer C without PMSF. The GST-fused ACO2 protein was eluted with buffer C containing 30 mM glutathione and pooled fractions were stored at −80 °C. His-ISCU was obtained from purification of the NFS1–ISD11–ISCU complex[47] and a further gel filtration step onto a Superdex-75 (Pharmacia) equilibrated with buffer D (100 mM Tris-HCl pH 7.5, 100 mM NaCl). *E. coli* apo-ferredoxin (Fdx) was expressed in Terrific broth for 18 h at 28 °C and purified in three steps using DEAE-cellulose, anion and hydrophobic columns. Apoferredoxin was prepared by boiling purified holoferredoxin in the presence of 100 mM EDTA and 500 mM DTT and desalting in buffer D. Protein concentrations were measured by Bradford assay using bovine serum albumin as a standard. FPLC gel filtration with an analytical Superdex-75 (Pharmacia Amersham Biotech) at a flow rate of 0.5 ml min$^{-1}$ equilibrated with buffer E (100 mM Tris-HCl pH 7.5, 100 mM NaCl, 5 mM DTT) was used for size determination of oligomerization state of the ISCA proteins. A gel filtration calibration kit (calibration protein II, Boehringer Inc) was used as molecular weight standards.

**Whole cell Mössbauer spectroscopy.** BL21(DE3) *E. coli* strain transformed with pET-ISCA1 or pET-ISCA2 or pACYC-ISCA2 plasmids were cultivated aerobically in $^{57}Fe$-enriched MOPS medium prepared as described[48]. Briefly, transformed bacteria were cultured overnight at 37 °C in LB medium supplemented with 100 µg ml$^{-1}$ ampicillin. Three ml were used to inoculate 2 × 1 l of MOPS supplemented with 100 µg ml$^{-1}$ ampicillin and 20 µM $^{57}FeCl_3$. Cells were incubated at 37 °C with agitation (200 rpm) until reaching an $OD_{600}$ of about 0.5–0.6. Protein expression was then induced by adding 0.5 mM IPTG during 3 h at 37 °C for ISCA2 or overnight at 20 °C for ISCA1. Under these conditions both proteins were expressed under a soluble form, although the amount varied for ISCA1 from one experiment to another. Control cultures were obtained without IPTG induction. Cells were harvested by centrifugation (4,000*g*, 20 min, 4 °C), washed twice with MOPS buffer containing 50 mM NaCl and 22 mM glucose and centrifuged again for 10 min at 5,000*g*, 4 °C. Cells were then packed in a Mössbauer cup (0.6–0.7 g of cells per Mössbauer cup) and frozen in liquid nitrogen before analysis. Mössbauer experiments were successfully repeated three times for ISCA1, four times for ISCA2, and twice for pACYC-ISCA2 in order to ensure reliable data and the presented spectra reflect the best experiment in terms of intensity and simulation. *In cellulo* Mössbauer spectra were recorded using a horizontal transmission 4 K closed cycle refrigerator system from Janis and SHI and a 100mCi source of $^{57}Co$(Rh) (ref. 49). All velocity scales and isomer shifts were referred to the metallic iron standard at room temperature. Analyses of the data were performed with the software WMOSS4 Mössbauer Spectral Analysis Software (www.wmoss.org, 2009–2015).

Simulations were first performed on the two control cell spectra. Four contributions were necessary to fully reproduce the experimental data. Introduction of a fifth contribution due to $[Fe_2S_2]^{2+}$ unavoidably led to low percentages (<4% for the whole cluster). The quality of the simulations is

significantly lost for Fe$_2$S$_2$ contributions larger than 4%. We constrained the nuclear parameters to be identical for the two calculations; only the linewidths and the relative area were allowed to differ. From the isomer shift value, two components were associated to HS Fe$^{II}$ species. Nuclear parameters $\delta = 0.45$ mm s$^{-1}$ and $\Delta E_Q = 1.15$ mm s$^{-1}$ were typical of $S = 0$ [Fe$_4$S$_4$]$^{2+}$ clusters and low-spin (LS) ferrous haem centres. A broad doublet with parameters $\delta = 0.48$ mm s$^{-1}$ and $\Delta E_Q = 0.61$ mm s$^{-1}$ is typical of Fe$^{III}$ (phosphate) oxyhydroxide nanoparticles (denoted Fe$^{III}$ NP)[50–52]. Simulations were then run to reproduce the two experimental spectra recorded on cells with overexpressed ISCA1 or ISCA2. The nuclear parameters (isomer shift and quadrupole splitting) for the four components identified in the control cell spectra were fixed to their determined values. Only the relative areas were allowed to vary. A fifth component was introduced owing to the presence of the line at $\approx 0.5$ mm s$^{-1}$ (see main text). Since we suspect the presence of [Fe$_2$S$_2$]$^{2+}$ clusters, we fixed the nuclear parameter values of the two Fe sites in a 1:1 ratio to that determined for $^{Nif}$IscA (ref. 53).

**Spectroscopic analyses on purified proteins.** UV-visible absorption spectra were recorded with an Uvikon XL spectrophotometer (BioTek Instruments). $^{57}$Fe-Mössbauer spectra on purified proteins were recorded using 400 μl cuvettes containing 0.38–1.1 mM protein. Spectra were recorded on a spectrometer operating in constant acceleration mode using an Oxford cryostat that allowed temperatures from 1.5 to 300 K and a $^{57}$Co source in rhodium. Isomer shifts are reported relative to metallic iron at room temperature. EPR spectra of purified proteins (250–500 μM) were obtained after reduction for 10 min with 1 mM dithionite prepared in buffer F (0.1M Tris-HCl pH 8) or for 40 min with 50 mM DTT prepared in buffer F. Spectra were recorded on a Bruker EMX (9.5 GHz) or ER200D EPR spectrometers equipped with an ESR 900 helium flow cryostat (Oxford Instruments). Double integrals of the EPR signals and spin concentration were obtained through the Win-EPR software using the spectrum of a 200 μM Cu(EDTA) standard recorded under non-saturating conditions.

**In vitro Fe–S cluster transfer.** For Fe–S transfer between ISCA and ISCU, FeS-ISCA (ISCA1 and ISCA2, 60 nmol) were incubated in buffer G (0.1 M Tris-HCL pH 8, 50 mM NaCl) under anaerobic conditions with one equivalent of apo-ISCU for 1 h. After separation onto an analytical Superdex-75 column, equilibrated in buffer G, each protein was analysed for its iron and sulfur content and its UV-vis. spectrum. The experiment was also performed the other way around, using FeS-ISCU and apo ISCA (ISCA1 and ISCA2) under the same conditions (incubation time, protein equivalent, and so on). ISCA apo-proteins were prepared by incubating as-purified proteins with 20 mM EDTA and 2 mM dithionite under anaerobic conditions for one night and desalting onto a NAP25 column equilibrated with buffer G. Iron content was measured to check that negligible iron is still bound to proteins.

For Fe–S transfer between ISCA and Apo-Fdx, purified Apo-Fdx (180 μM) was incubated anaerobically in 200 μl buffer H (0.1 M Tris-HCl pH: 8; 50 mM KCl) with native ISCA1 (0.9 iron and 0.9 sulfur atom/monomer) or ISCA2 (1 iron and 0.9 sulfur atom/monomer) in order to provide 2 iron and sulfur atoms/ferredoxin. Fe–S transfer was followed by monitoring the UV-visible absorption in the 300–600 nm region and by EPR analysis of the ISCA-Fdx protein mixture after reduction with 2 mM dithionite.

To assess mouse aconitase activation, apo-ACO2 (0.2 nmol) was pretreated with DTT and then incubated anaerobically in a glove box at 18 °C in buffer I (50 mM Tris-HCl pH: 7.6) with a tenfold molar excess of the native ISCA1 (0.9 iron and sulfur atom/monomer) or ISCA2 (0.6 iron and 0.7 sulfur atom/monomer) in order to provide 4 iron and 4 sulfur atoms per ACO2. After 5, 10, 15, 20 and 30 min incubation, aconitase activity was assessed as described[54]. Briefly, ACO2-ISCA mixtures were added to 0.6 mM MnCl$_2$, 25 mM citrate, 0.5 U isocitric dehydrogenase, 0.25 mM NADP$^+$, 50 mM Tris-HCl, pH 7.6, in a 100 μl final volume and NADPH formation was monitored at 340 nm. The 100% activity corresponds to the activity of the chemically reconstituted ACO2 prepared by incubating apo-ACO2 with 4 molar excess of ferrous iron and sulfur for 30 min in the presence of 5 mM DTT (4 μmol min$^{-1}$ mg$^{-1}$).

**Cell cultures.** HeLa cells (ATCC-CCL-2) and Neuro-2a (N2a)(ATCC-CCL-131) cells were cultured in DMEM medium supplemented with 1 g l$^{-1}$ glucose, 5% fetal calf serum and 40 μg ml$^{-1}$ gentamycin at 37% with 5% CO$_2$. Cell lines tested negative for mycoplasma contamination. HeLa cells and N2a cells were transfected with the different pcDNA3.1 constructs (Isca1, Isca2, Iba57, Glrx5, Fdx2) using Fugene 6 transfection reagent (Roche) and Lipofectamine transfection reagent (Thermofisher), respectively, following the manufacturer's recommendations. Transfected cells were harvested 24–48 h after transfection.

Primary sensory neurons were obtained from the dissection of dorsal root ganglia (DRGs) of E13.5 mouse embryos[55]. Briefly, DRGs were collected and digested in 1 ml of 2.5‰ trypsin for 45 min at 37 °C. Digestion was stopped by adding two volumes of C-medium (MEM, glucose 4 g l$^{-1}$, 10% fetal bovine serum, glutamine 2 mM, NGF 50 ng ml$^{-1}$) supplemented with penicillin and streptomycin (P/S), and 1 volume of fetal bovine serum. Cells were centrifuged at 800g, 10 min at RT and mechanically dissociated in C-medium + P/S using a 1 ml micropipette. For western blot analysis, cells were plated on poly-lysine/collagen-coated 35 mm plates in C-medium + P/S (at a concentration of about 30 DRGs per plate). For Seahorse analysis, the cell suspension was complemented with matrigel (Corning) in a 1:19 ratio and with 10 μg ml$^{-1}$ of laminin (Sigma) to a final concentration of approximately 0.03 DRG per μl. Eighty μl per well were then plated on poly-lysine/laminin-coated XF96 cell culture microplates (Seahorse bioscience). The day after dissection medium was changed to C-medium supplemented with 10 μM fluorodeoxyuridine + 10 μM uridine (FUdR) to get rid of dividing cells (treatment was repeated the day after transduction with rAAV for cultures on 35 mm plates). The culture medium was then changed every 2 days with NB medium (Neurobasal, glucose 4 g l$^{-1}$, B27, glutamine 2 mM, NGF 50 ng ml$^{-1}$). Cultures were maintained at 37% with 5% CO$_2$. Transduction of neurons with the different rAAV2/9 was carried out overnight using a concentration of $9 \times 10^9$ vg ml$^{-1}$ in C-medium (using 1 ml for 35 mm plates and 80 μl per well for the XF96 microplate). For 35 mm plates, infection was performed 4 days after dissection, whereas infection was performed the night after dissection for XF96 microplates. Cells were collected 15 days after infection and pellets were kept at $-80$ °C.

**Immunoprecipitation.** IPs were carried out using mitochondrial-enriched protein extracts from HeLa or N2a cells overexpressing FLAG-tagged proteins[56]. Cells were collected in PBS and mitochondrial-enriched fractions were obtained by incubating cell pellets in Tris-HCl 100 mM, pH 7.5, 10% glycerol supplemented with 0.014% digitonin at 4 °C for 10 min, and by centrifuging the suspension 10,000g 10 min at 4 °C. The resulting pellet was resuspended in Tris-HCl 10 mM pH 7.4, 10% glycerol, 100 mM NaCl, 50 mM KCl, Complete protease inhibitor cocktail (Roche), 0.2% Triton X-100, incubated on ice 20 min and centrifuged 10,000g 10 min at 4 °C. The resulting supernatant corresponding to the mitochondria-enriched protein extracts (0.3–8 μg) were incubated with 25–100 μl FLAG M2 coupled resin (SIGMA) (50% slurry, prepared as recommended by the manufacturer) in the presence of Complete protease inhibitor (Roche). Suspensions were incubated 2 h at 4 °C, centrifuged 10 min at 12,000g and beads were washed twice with 1 ml PBS. Elution was performed by incubating the resin 5 min with glycine buffer (0.1 M glycine, pH 2.8) and centrifugation for 10 min at 12,000g.

**Mass spectrometry analysis.** Immunoprecipitated samples were analysed by MudPIT as follows (three biological replicates for each individual protein). Flag IP eluates were TCA-precipitated, urea-denatured, reduced, alkylated and digested with endoproteinase Lys-C (Roche), followed by modified trypsin digestion (Promega). Peptide mixtures were loaded onto a triphasic 100 μm inner diameter fused silica microcapillary column, packed with C18 reverse phase (Aqua, Phenomenex) and strong cation exchange (Partisphere SCX, Whatman) particles. Loaded columns were placed in-line with a Dionex Ultimate 3,000 nano-LC (Thermo Fisher Scientific) and a LTQ Velos linear ion trap mass spectrometer equipped with a nano-LC electrospray ionization source (Thermo Fisher Scientific). A fully automated 12-step MudPIT run was performed as previously described[57], during which each full MS scan (from 300 to 1,700 m/z range) was followed by 20 MS/MS events using data-dependent acquisition. Proteins were identified by database searching using SEQUEST HT within Proteome Discoverer 1.4 software (Thermo Fisher Scientific) against a Homo sapiens database containing 20,206 protein sequences (Swissprot release from 2012-10-05) in which protein sequences of ISCA1, ISCA2, IBA57, FDX2 or GLRX5 have been replaced by the corresponding mouse protein sequence fused to Flag tag. Cysteine residues were considered to be fully carbamidomethylated (+57 Da statically added) and methionine considered to be oxidized (+16 Da dynamically added). Peptides were filtered with XCorr values equal to 1.5 (+1), 2.5 (+2), 3.0 (+3) and 3.2 (>+3). Only peptides with a minimum length of seven residues and maximum DCn values of 0.1 were retained. Specific interacting partners were considered to be enriched 10× over negative controls and mitochondrial input samples. IP normalization was performed using at least one control IP and two mitochondrial input samples. Remaining proteins were selected for mitochondrial localization using SWISSprot library. SAF (PSM/protein length) and NSAF (SAF(protein)/sum SAF (all proteins)) values were calculated from sum PSM values of replicate IP experiments[57]. NSAF × 100 values were used for representation.

**Mice.** All mice were in 100% C57BL/6J background. Mice were maintained in a temperature and humidity-controlled animal facility, with a 12-h light–dark cycle and free access to water and a standard rodent chow (D03, SAFE). Both male and female mice were used in all experiments. All animal procedures and experiments were approved by the local ethical committee (Comité d'Ethique en Expérimentation Animale IGBMC-ICS) and the Ministère de l'Education Nationale et de l'Enseignement Supérieur et de la Recherche (project #02406.03). For knockdown experiments, 4-week-old C57BL/6J wild-type mice were anaesthetized by intra-peritoneal injection of ketamine-xylazine (75 and 10 mg per kg body weight, respectively) and a dose of $2.5 \times 10^{10}$ vg in a total volume of 25 μl of rAAV2/1 was administered in each TA. Recombinant AAV2/1 expressing the specific shRNAs and the rAAV2/1 with the scrambled shRNA were injected at contralateral sides, respectively. Three or 6 weeks post-infection, mice were anaesthetized by intra-peritoneal injection of ketamine-xylazine, dissected and killed. All experiments have biological replicates ($n = 5$–10) for 3 w.p.i. and $n = 3$ for 6 w.p.i. The sample

size of a minimum of three biological replicates was estimated based on the inter-individual biological variability and the variability of injection. The samples were not randomized since the contralateral side injected with the scrambled shRNA served as the control for each experiment. For molecular analysis, tissue samples were directly snap-frozen in liquid nitrogen and kept at −80 °C. For histology, skeletal muscle were embedded in OCT Tissue Tek (Sakura Finetechnical, Torrance, CA, USA) and snap-frozen in isopentane chilled in liquid nitrogen. For electron microscopy, tissue samples were fixed in 2.5% paraformaldehyde and 2.5% glutaraldehyde in cacodylate buffer (0.1 M, pH 7.2), post-fixed in 0.1 M cacodylate buffer supplemented with 1% osmium tetroxide for 1 h at 4 °C, dehydrated and embedded in Epon.

**Western blot and enzymatic measurements.** Tissues were homogenized using the ULTRA-TURRAX T25 basic (IKA-WERKE) in SDS-buffer (10 µl per 1 mg tissue; 280 mM Tris, 43% glycerol, 10% SDS, pH 6.8). Primary neurons were processed using TGEK buffer (10 mM Tris-HCl, pH 7.4, 1 mM EDTA, 10% glycerol, 50 mM KCl) supplemented with 0.2% Triton X100 and Complete inhibitor cocktail (Roche). After homogenization, samples were centrifuged at 12,000g, 10 min, 4 °C and the protein concentration of supernatants was determined using Bradford assay (Bio-Rad). Extracts were further processed using the SDS buffer to resuspend membrane proteins. Unless otherwise indicated, protein extracts were further denatured in loading buffer and loaded on Tris-glycine polyacrylamide gel[58]. Antibodies were incubated overnight at 4 °C in PBS-Tween or TBS-Tween (0.05%) using antibodies listed in Supplementary Table 3. Western blot was revealed using Amersham Hyperfilm ECL or Amersham Imager 600 (GE Healthcare). All original western blots can be found in Supplementary Fig. 4. SDH activities were measured spectrophotometrically by following the reduction of dichlorophenol indophenol (DCPIP) at 600 nm (ref. 59).

**Quantitative real-time PCR.** Total RNA was obtained from mouse tissues using Tri Reagent (Molecular Research Centre, Inc.) according to the manufacturer's protocol. Reverse transcription was carried out using Superscript II kit (Invitrogen). Quantitative RT–PCR was performed on a LightCycler 480 apparatus (Roche) using specific oligonucleotides, listed in Supplementary Table 4. *Hprt* expression was used as control.

**Histology and electron microscopy.** For histological analysis, 10 µm cryostat sections were stained with haematoxylin and eosin, or for the activity of SDH[59]. An incubation time of 10 min for SDH at 37 °C was required for the appearance of the nitroblue-diformozan precipitate before mounting. For electron microscopy, ultrathin sections (70 nm) of skeletal muscle were contrasted with uranyl acetate and lead citrate and examined with a Morgagni 268D electron microscope[59].

**AAV production and purification.** AAV2/1 and AAV2/9 vectors were generated by a triple transfection of AAV-293 cell line with the different pAAV constructs, pHelper and pXR1 or pAAV2_9 PENN P0008 (PENN Vector Core) for AAV serotype 1 or AAV serotype 9, respectively. Cell lysates were subjected to three freeze/thaw cycles, then treated with 50 U ml⁻¹ of Benzonase (Sigma) for 30 min at 37 °C, and clarified by centrifugation. Viral vectors were purified by Iodixanol gradient ultracentrifugation followed by dialysis and concentration against Dulbecco's Phosphate Buffered Saline using centrifugal filters (Amicon Ultra-15 Centrifugal Filter Devices 30 K, Millipore, Bedford, MA, USA). Physical particles were quantified by real-time PCR using a plasmid standard pAAV-eGFP, and titers were expressed as viral genomes per milliliter (vg ml⁻¹). All rAAVs were obtained with titers over $10^{12}$ vg ml⁻¹.

**Seahorse analysis.** Seahorse analyses were carried on a XFe96 extracellular flux analyser using the XF Cell Mito Stress Test Kit (Seahorse bioscience), following the manufacturer's recommendations and protocol. Experiments were performed in XF Base medium supplemented with 10 mM glucose, 2 mM pyruvate and 2 mM glutamine, pH 7.4, using 2 µM oligomycin, 0.5 µM FCCP and 0.5 µM of rotenone and antimycin. Data were analysed using the XF Cell Mito Stress Test Report Generator V2 (http://www.seahorsebio.com/support/software/stress-test-generator.php) and by calculating the basal respiration and spare respiratory capacity for each well.

**Antibody production and purification.** Polyclonal rabbit antibodies against mouse ISCA2 or IBA57 protein were produced using the GST-ISCA2 recombinant protein or the CLGDLQDYHKYRYQQG peptide, respectively. The peptide was coupled to activated KLH using Imject Maleimide Activated Carrier Protein Spin Kits (Thermo Scientific) in Tris-EDTA buffer (50 mM Tris, pH 8.5, 5 mM EDTA). Rabbits were injected with a 1:1 solution of the respective antigen and Freunds Complet adjuvant. Sera were purified using SulfoLink Coupling Resin (Thermo Scientific), according to the manufacturer's protocol, using His-ISCA2 or the respective epitope-peptide. Antibodies were eluted in acidic conditions (0.1 M glycine, pH 2.8). Antibody-containing fractions were then dialysed against PBS and stored in glycerol 29% with NaN₃ at −20 °C.

**Statistical analysis.** Differences between mean values were evaluated using the bilateral Student's *t*-test. $p < 0.05$ was considered significant.

**Data availability.** Data supporting the findings of this study are available within the article and its supplementary information files and from the corresponding author on reasonable request. The genes with *Ensembl* accession codes ENSMUSG00000044792 (Isca11), ENSMUSG00000021241 (Isca2), ENSMUSG00000021102 (Glrx5), ENSMUST00000215338 (Fdx2) and ENSMUSG00000049287 (Iba57) were used in this work.

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

## Acknowledgements

The authors thank Jean-Marc Latour for helpful discussions. We thank Françoise Piguet and Olivier Griso for contributions to primary neuronal cultures, Laurence Reutenauer for technical help, Belinda Cowling and members of Laporte's team for the set up of AAV intramuscular injection and helpful discussions. We also acknowledge the Ingestem platform and all the IGBMC platforms for their contributions and help. This work was supported by grants from the Friedreich Ataxia Research Alliance to A.M.; The Agence Nationale pour la Recherche with the ANR FRATISCA (ANR-14-CE09-0026) to S.O.d.C, G.B. and H.P.; the ANR FeStreS (ANR-11-BSV3-022-02) to S.O.d.C.; the Labex ARCANE (ANR-11-LABX-0003-01) to S.O.d.C. and G.B.; the Labex (ANR-10-LABX-0030-INRT) and Idex (ANR-10-IDEX-0002-02), as well as the European Community under the European Research Council (206634/ISCATAXIA) to H.P.

## Author contributions

L.K.B., S.O.d.C, I.S., M.-A.H., M.C., S.S., A.E., A.W., P.K., N.M. and A.M. performed experiments, M.F., S.O.d.C, G.B., H.P. and A.M. designed experiments, L.K.B., S.O.d.C, M.F., I.S., M.C., G.B., H.P. and A.M. analysed data, L.K.B., S.O.d.C, G.B., H.P. and A.M. wrote the manuscript.

## Additional information

**Competing interests:** The authors declare no competing financial interests.

