## [Peer Review File · Nature Communications]

Reviewer #1 (Remarks to the Author)

The authors have attempted to resolve questions about the roles of ISCA1 and ISCA2 in mammalian iron-sulfur cluster biogenesis. They begin by over-expressing each protein in bacteria, and they detect 2Fe2S clusters. These results do not necessarily imply what type of FeS cluster will be found in vivo. Similar experiments have been conducted before, but these were performed differently. Nevertheless, their results have limited implications for in vivo settings. Next, they use mass spec and coIP results to suggest that ISCA1 and ISCA2 participate in different networks. Upon AAV infection with siRNA, they find that knockdown of ISCA1 results in diminished amounts of various FeS proteins by western blot. The only activity assay is the SDH stain of figure 4B, but many of the cells in this figure show lack of SDH activity, not only in ISCA1 knockdown, but also ISCA2 knockdown. These results would be strengthened by performing activity assays on muscle lysates. They should also perform mitochondrial aconitase activity assays on lysates from muscle lysates, and also from their infected primary sensory neurons.

This is the first report of a mammalian phenotype for ISCA loss of function, but the phenotyping is incomplete, and in some respects, not technically solid (Figure 5, C and D).

The conclusions that ISCA1 has a role that ISCA2 lacks is rather preliminary, but could be strengthened with more experimentation and better replicates.

The analysis of the FeS of ISCA from bacterial overexpression would be better in a separate paper. Here, the efforts to characterize the phenotype in muscle is the most novel part.

The narrative does not present a cohesive picture. The mass spec data is interesting, but they excluded "non FeS proteins" and set a threshold for reporting that may have led to some over-interpretation. It would be better to report the whole data set in a supplementary figure, and include their annotated results in a main figure.

The paper is not clear because it mixes in vitro and in vivo results, and the in vivo results lack important experiments.

Reviewer #2 (Remarks to the Author)

This manuscript reports on the biochemical/biological functions of ISCA1 and ISCA2 using mouse as the experimental system. A variety of interesting observations are made, including aspects of in vitro cluster formation and trafficking, and the effect of depletion of either ISCA1, or ISCA2. This reviewer did not notice any particular deficiency in the execution of the experiments and therefore will concentrate on the conclusions. Although the results do appear to have been performed adequately there are some rather striking conclusions that are made, some of which are justified and some, perhaps, not. The authors have been able to separately and heterologously produce his-tagged versions of ISCA1 or ISCA2 in Escherichia coli and provide convincing evidence that the forms produced in this way carry 2Fe-2S clusters. This is an interesting finding that contrasts with what others have reported in the literature. Namely, others have reported that similar heterologous production of ISCA forms from other organisms, including bacteria, yeast, and other higher eukaryotes do not contain clusters but Fe-S species can be reconstituted on the purified protein using Fe and S. Also, others have reported being able to heterologously produce ISCA that contains Fe but no sulfur. Yet, the authors make the sweeping conclusion that they have demonstrated, based on their results alone, that ISCA is not an Fe carrier. They have produced no data to back this statement up. There are probably hundreds of different examples in the literature that show that certain proteins have more than one function and/or have different functions under different physiological conditions. The authors report that ISCA1 and ISCA2 behave as Fe-S carriers in vitro. They do, in fact present data to support that statement. However, as mentioned in the criticism above, they have presented no data that ISCA cannot also function as a scaffold protein. Their conclusion is based on their observation IscU can transfer an Fe-S cluster to ISCA but the reverse transfer cannot happen. That only means that uni-directional transfer occurs under their experimental conditions. It has been reported many times that incubation of ISCA with ISCS, cysteine and Fe leads to formation of clusters on ISCA in vitro. So it is clear that ISCA can, in fact serve as a scaffold for cluster synthesis in vitro. The point here is not that the authors have incorrectly interpreted their data, their hypothesis is reasonable. However, there is an issue that

they present their interpretation as an established fact, when that is not the case. If ISCA proteins serve as independent scaffolds it is not necessarily expected that transfer of its clusters to a different scaffold would necessarily need to happen. The above comments are relatively minor. However, the interpretation made of the phenotypes and identification of interacting partners is a more serious concern. In the interpretation of their work the authors state, or at least strongly imply, that the work of Shetel et al is flawed. The work of Shetel et al reported on the fact that a strong phenotype can be recognized in HeLa cells when either ISCA1 or ISCA2 is depleted and that the two proteins have the capacity to form a complex in vivo. Complex formation was also shown in the present work, also using a HeLa-based experimental system, but when using an entirely different experimental system the authors of the present work recognize a phenotype when ISCA1 is depleted but not when ISCA2 is depleted. From this they make the conclusion that "our results strongly suggest that HeLa cells or dividing cells may not be reliable to investigate the role of the proteins in physiological conditions and to decipher the molecular mechanisms driving the late mitochondrial Fe-S biogenesis". Exactly the reverse conclusion could also be made, after all Shetel et al do report a phenotype and it seems that a system for which a phenotype can be tracked would be a superior experimental tool. These comments are not intended to be hostile. Instead this reviewer, who is not a scientific competitor, believes that the overall tone of the manuscript, in its present form, does a disservice to the authors. In the abstract the authors state "collectively, our data question the current view of late Fe-S mitochondrial machinery" although what, exactly is being questioned is not clear. There are indeed some very nice aspects that are reported and highly merit publication in a premier journal. For example, there is strong evidence that ISCA1 and ISCA2 are likely to have some different interacting partners. However, as confirmed in the work ISCA1 and ISCA2 also interact with each other in vivo. This strongly suggests a dynamic network, together with nice biochemical work, suggesting that ISCA forms are likely to harbor Fe-S clusters (although only shown in a heterologous system) and that ISCA proteins could be (not proven) acting as intermediate Fe-S carriers. It is unfortunate that, rather than concentrating on the novel observations, which underscore the complexity of the process, the authors choose to concentrate on emphasizing differences of their work with the published literature with an implicit suggestion that other approaches and suggestions are inherently flawed. The title of the manuscript is misleading. It is not proven that "ISCA1 governs mitochondrial 4Fe-4S cluster biogenesis" as claimed. Rather it shows that mitochondrial 4Fe-4S cluster biogenesis can occur in the absence of ISCA2. It would be more appropriate to indicate that ISCA1 is essential for 4Fe-4S cluster synthesis and ISCA2 is not. After all, it could well be that the governing species is an ISCA1/ISCA2 hybrid and in the absence of ISCA2 its function can be replaced, in part or whole, by an ISCA1 X 2 species. Also, the claim about "normal physiological conditions" would appear to be inappropriate given that the authors are using transfected cells that have shut down one or the other ISCA species. Can this really be considered normal physiological conditions. If ISCA is strictly a 2Fe-2S cluster carrier protein how is it that when it is produced heterologously in *E. coli*, where the native primary biosynthetic machinery is clearly not present it contains a 2Fe-2S cluster. It must be assumed in this situation that cluster-loaded IscU from *Escherichia coli* is the source of the cluster but this possibility is not explored in the present work. "Evolutionary" should probably be "Evolutionarily". Line 47 "consisting in" should probably be "consisting of".

Reviewer #3 (Remarks to the Author)

Overview: Overall, this is a significant manuscript that may eventually be acceptable for publication. The authors have made significant advances in understanding the function of the IscA type protein that are associated with Fe/S cluster assembly. Mammals contain two IscA homologs, Isca1 and Isca2. Prior to this paper, the two proteins were viewed as being redundant, but the authors show that only Isca1 plays the role in Fe/S cluster assembly. They also show that both proteins bind Fe₂S₂ clusters and can transfer clusters. The experiments are generally presented well and the conclusions are reasonable. That being said, I do have some critical comments that must be addressed before I could recommend this paper for publication. I felt that the spectroscopy portion of the paper needs to be bolstered.

Comment 1: In Figure 1, whole cell spectra in C of the overexpression mutants show a small feature which is interpreted as the high-energy line of a quadrupole doublet due to $[2\text{Fe-2S}]_2^+$ clusters. However, the spectral intensity of this doublet represents only 10 - 12% of the total; most spectral intensity is due to high-spin Fe(II) and unresolved features in the middle of the spectrum that are attributed to Fe(III) nanoparticles. A lot of interpretive "weight" is being placed on a minor spectral feature, which is uncomfortable. How many replicate spectra have been obtained, and how reproducible is this minor spectral feature? If the experiment has only been done once, or if the presented spectra are the best that they have collected, this must be repeated. If they have done the experiment multiple times, and see the same features, this should be mentioned in the text as it would strengthen their argument.

Comment 2: A related concern for the whole-cell spectra of Figure 1C is that the control spectra are significantly different from each other, with one showing substantial nanoparticles and the other showing less. Why? Again this is a bit unsettling. How many replicate spectra were obtained, and are these differences reproducible? The state at which the cells were harvested (log or post-log) could make a difference in nanoparticle formation.

Comment 3: In Table 1, the % of $[\text{Fe}_2\text{S}_2]_2^+$ cluster in the control whole-cell samples is stated to be "0", but it must be possible to include some small percentage in a spectral simulation without a noticeable reduction in fit quality (my guess is that about 5% would fit in this way). Thus, it would be more accurate to state in the table something like "< 5%" rather than "0". But in this case, there is not much difference between control and overexpression spectra (e.g. 4% vs 11%).

Comment 4: I believe that the Figure 1 legend should say "applied parallel to the γ beam"

Comment 5: The authors report that they were unable to reduce the $[\text{Fe}_2\text{S}_2]_2^+$ clusters more than 15% - 20%. Hmmm. Have they tried this more than once? Have they tried dithionite at high pH? (I don't think the methods section states the pH of the dithionite solution. Dithionite is a more powerful reductant at high pH like 8.5). Did they examine the Mossbauer spectrum of the sample after treatment with dithionite? Was the remaining 80% - 85% of the cluster still intact and in the $[\text{Fe}_2\text{S}_2]_2^+$ state (or was there degradation? Or formation of magnetic material that might indicate a higher spin state??).

Comment 6: On page 6, line 124, I found it confusing that they mentioned that 78 K Mossbauer spectrum was "consistent with Fe(III) ($S = 5/2$) in tetrahedral environment..." but not to mention that it was also consistent with $[\text{Fe}_2\text{S}_2]_2^+$ clusters, which ended up supported by the 4.2 K data. They say that the parameters "unambiguously demonstrate" the presence of a $[\text{Fe}_2\text{S}_2]_2^+$ cluster with $S = 0$, but that seems too strong without any high-field data and simulation fit assuming diamagnetism.

Comment 7: On page 7, the authors note that "in bacterial cells that overexpress Fe-S proteins, the iron content under the form of Fe-S never exceeded 20-40% of total iron." Is this a general "rule" or only the result found in the two cited reports? I've not heard of this before.

Comment 8: The spectra in Figure 2 are somewhat confusing. In the text, they discuss mixing Holo-A1 and holo-A2 with apo-ISCU, and I expected that the corresponding spectra would be on the left (however they are on the right). More importantly, the text says that there was no cluster transfer but the data seem to show a 2-fold drop in the absorbance due to holo-ISCA2 during the experiment, and an increase in the absorption due to ISCU. Doesn't this suggest that a portion of the clusters were indeed transferred?

Comment 9: Perhaps I missed it but I was unable to find an explanation of how the apo-forms of these proteins were prepared.

Comment 10: Have the authors monitored reduction of the clusters by UV-vis spectroscopy, using dithionite as the reductant? Can they determine whether the redox potential of the cluster is too low or whether reduction is kinetically sluggish? Why are they seeing partial reduction?

Comment 11: In Figure 4, they seem to have repeated the Western analysis for a given column, but the figure legend doesn't mention this. Please clarify.

Comment 12: On page 13, the authors state "...but could shield a Fe₂S₂..." I was a bit confused regarding the word "shield". (bind??)

Comment 13: There was a confusing shift in the paper and the authors need to reconcile and clarify this. In the first part of the paper, Isca1 and Isca2 proteins were both characterized and it seems that the lesson taught is that both are "viable" as Fe-S cluster carrier proteins, accepting a cluster from Iscu and donating it to a Fe₂S₂ ferredoxin. In fact, the evidence for Isca2 functioning in this regard is stronger than for Isca1. Both Fe/S loaded proteins could activate apo-aconitase implying the assembly of an Fe₄S₄ cluster. However, when the function of these proteins are investigated in mice and cell culture, Isca1 was found to be involved in mitochondrial Fe/S cluster assembly while the role of Isca2 is undefined but not involved in these processes. This apparent incongruency needs to be resolved or at least better clarified and explained by the authors. Left as is, this issue will cause confusion and it detracts from the impact of the paper.

We thank the reviewers for carefully considering our work. Their suggestions were helpful, and we feel that the resulting changes significantly improved the manuscript. We have revised the manuscript accordingly and the major changes are outlined here:

- We have revised the discussion and the manuscript throughout changing the general tone/critics to answer to reviewer 2 which appreciated the work, but was destabilized by the general tone of the manuscript.
- We provide further explanation and have added SDH enzymatic activities and the full dataset from the mass spectrometry analysis to address reviewer 1's concerns.
- We provide further information and performed additional experiments to answer questions of reviewer 3.

Our point-by-point response to each reviewer can be found below.

Reviewer #1 :

The authors have attempted to resolve questions about the roles of ISCA1 and ISCA2 in mammalian iron-sulfur cluster biogenesis. They begin by over-expressing each protein in bacteria, and they detect 2Fe2S clusters. These results do not necessarily imply what type of FeS cluster will be found in vivo. Similar experiments have been conducted before, but these were performed differently. Nevertheless, their results have limited implications for in vivo settings.

Response:

We agree with reviewer 1 on the fact that characterization of ISCA proteins *in vitro* does not inform us on the type of cluster present *in vivo*. However, these *in vitro* data become important when combined with the Mössbauer experiments performed *in cellulo*. Indeed, combined together, our data show that ISCA proteins are Fe₂-S₂ proteins within the cells and that the observation of a Fe₂-S₂ cluster after purification is not due to a degradation of a Fe₄-S₄ cluster during purification, an important information.

Upon AAV infection with siRNA, they find that knockdown of ISCA1 results in diminished amounts of various FeS proteins by western blot. The only activity assay is the SDH stain of figure 4B, but many of the cells in this figure show lack of SDH activity, not only in ISCA1 knockdown, but also ISCA2 knockdown. These results would be strengthened by performing activity assays on muscle lysates. They should also perform mitochondrial aconitase activity assays on lysates from muscle lysates, and also from their infected primary sensory neurons.

Response (including a new figure panel added to the manuscript):

We would like to remind the reviewer that skeletal muscle is composed of mixed fiber-types that have different metabolic state i.e. different mitochondrial content and different levels of SDH activity (for example see: Helge Amthor et al. PNAS 2007;104:1835-1840; Sundaram et al. J. Clinical Neuroscience 2011: 18: 535-538; Sugimoto et al., Brain and Development 2000: 22:158-162). Therefore, in Figure 4B, the patchy staining that is seen in the control and ISCA2 knockdown is a normal SDH stain of a mixed skeletal muscle cross-section, the dark fibers corresponding to fibers with high oxidative capacity. However, in the ISCA1 or ISCU knockdown, a major loss of blue colored nitroblue-diformozan precipitate, which corresponds to SDH activity, is seen. It is therefore possible to draw the conclusion that SDH

activity is decreased in the ISCA1 and ISCU knockdown, but not in the ISCA2 knockdown. We however agree with the reviewer that this is a qualitative assessment and not a quantitative assessment. We therefore, as suggested by the reviewer, measured SDH activity in tibialis anterior (TA) muscle lysates from CTL, ISCA1, ISCA2 and ISCU knockdowns using a spectrophotometric method (Supplemental Fig. 2C). The results demonstrate that there is a clear SDH activity deficit in the TA muscle lysate from ISCA1 and ISCU knockdown, while there is no significant loss of SDH activity in the ISCA2 knockdown, confirming the original results on muscle cryosections (Fig. 4B).

While we agree that quantitatively measuring a second activity of an Fe-S enzyme, such as aconitase, would further strengthen the results, the amount of material in a TA muscle is very limited and prevents proper evaluation of aconitase activity, an enzyme that is extremely sensitive to oxygen. Several attempts at measuring aconitase activity were performed, but were inconclusive, with very low activity and high variability in different wild-type samples, despite the expertise being present in the laboratory (see for example Martelli et al 2015 Cell Metabolism).

However, the decrease in lipoic acid (LA) bound to pyruvate dehydrogenase and α -ketoglutarate dehydrogenase complexes also reflects a decrease in lipoic acid synthase activity, therefore further supporting the results.

The amount of material in primary sensory neurons is even more limited than TA muscle, and therefore it is difficult to measure enzymatic activity using the standard spectrophotometric methods. However, using Seahorse Analyzer on live neuronal cultures, we clearly demonstrated that mitochondrial respiratory function (basal respiration and spare capacity) is affected in ISCA1 knockdown and not in ISCA2 knockdown (Fig. 5E). These results demonstrate that ISCA1 and not ISCA2 is essential for mitochondrial respiratory function under the present conditions.

Altogether, we hope that the addition of the SDH activity from muscle lysate, and our present explanation, will be sufficient to lift the reviewer's concern.

This is the first report of a mammalian phenotype for ISCA loss of function, but the phenotyping is incomplete, and in some respects, not technically solid (Figure 5, C and D). The conclusions that ISCA1 has a role that ISCA2 lacks is rather preliminary, but could be strengthened with more experimentation and better replicates.

Response:

We understand that part of the concern of the reviewer is that it was unclear from the original submitted manuscript how many replicates were present for each experiment. We have therefore clarified this throughout the manuscript.

Figure 5C and D are representative western blot from proteins lysates from sensory neuronal cultures of two independent samples. In total, 6 independent experiments have been performed, and equivalent results were obtained.

Regarding the replicates in the mouse, all *in vivo* experiments have been performed in numerous biological replicates (n=3-10) (we have added this information in the methods section as well as in the figure legends). For all histological sections or RT-PCR experiments 3 weeks post infection (w.p.i.), there are 5 TA samples/conditions. For all western blot analysis, 5-10 TA samples for each condition were analyzed. For all experiments at 6 w.p.i., there are 3 biological replicates. The data presented throughout the paper are representative western blots, which also take into account the variability that can be seen from one animal to another.

The narrative does not present a cohesive picture. The mass spec data is interesting, but they excluded "non FeS proteins" and set a threshold for reporting that may have led to some over-interpretation. It would be better to report the whole data set in a supplementary figure, and include their annotated results in a main figure.

Response:

We understand the concern of the reviewer for not reporting the full dataset from the Mass spectrometry analysis, and therefore, we have provided in Supplementary Table 2A the full list of proteins identified in each dataset and non-filtered, as requested by the reviewer, and changed the text in the manuscript accordingly. However, in order to analyze Mass Spectrometry data it is necessary to choose a threshold and/or to stratify according to pathways of interest. Here, we have taken a high stringency criterion (ten-fold enrichment), and we have decided to look at primary partners within the Fe-S machinery, based on literature and potential function of the proteins. Going beyond Fe-S would decrease the cohesion of the manuscript and would lead to speculations and over-interpretation.

The analysis of the FeS of ISCA from bacterial overexpression would be better in a separate paper. Here, the efforts to characterize the phenotype in muscle is the most novel part.

The paper is not clear because it mixes in vitro and in vivo results, and the in vivo results lack important experiments

Response:

We disagree with the reviewer. We believe that interdisciplinary aspect of the manuscript is a strength, because it is important to correlate biochemical data with *in vivo* characterization to better understand the complex process involved in Fe-S biosynthesis.

Reviewer #2:

This reviewer did not notice any particular deficiency in the execution of the experiments and therefore will concentrate on the conclusions. Although the results do appear to have been performed adequately there are some rather striking conclusions that are made, some of which are justified and some, perhaps, not. The authors have been able to separately and heterologously produce his-tagged versions of ISCA1 or ISCA2 in *Escherichia coli* and provide convincing evidence that the forms produced in this way carry 2Fe-2S clusters. This is an interesting finding that contrasts with what others have reported in the literature. Namely, others have reported that similar heterologous production of ISCA forms from other organisms, including bacteria, yeast, and other higher eukaryotes do not contain clusters but Fe-S species can be reconstituted on the purified protein using Fe and S. Also, others have reported being able to heterologously produce ISCA that contains Fe but no sulfur. Yet, the authors make the sweeping conclusion that they have demonstrated, based on their results alone, that ISCA is not an Fe carrier. They have produced no data to back this statement up. There are probably hundreds of different examples in the literature that show that certain proteins have more than one function and/or have different functions under different physiological conditions. The authors report that ISCA1 and ISCA2 behave as Fe-S carriers in vitro. They do, in fact present data to support that statement. However, as mentioned in the criticism above, they have presented no data that ISCA cannot also function as a scaffold protein. Their conclusion is based on their observation IscU can transfer an Fe-S cluster to ISCA but the reverse transfer cannot happen. That only means that uni-directional transfer occurs under their experimental conditions. It has been reported many times that incubation of ISCA with ISCS, cysteine and Fe leads to formation of clusters on ISCA in vitro. So it is clear that ISCA can, in fact serve as a scaffold for cluster synthesis in vitro. The point here is

not that the authors have incorrectly interpreted their data, their hypothesis is reasonable. However, there is an issue that they present their interpretation as an established fact, when that is not the case. If ISCA proteins serve as independent scaffolds it is not necessarily expected that transfer of its clusters to a different scaffold would necessarily need to happen. The above comments are relatively minor.

Response:

We thank the reviewer for the positive feedback on the execution of the experiment and the adequacy of the data presented in the manuscript. The reviewer is correct, we have not excluded that ISCA proteins could be Fe carriers under different conditions, and we have corrected the text accordingly. The fact that IscA can assemble an Fe-S cluster in the presence of a cysteine desulfurase, iron and cysteine does not prove that this protein is a scaffold. The above experimental works with apo-aconitase or any other apo-proteins that require an Fe-S cluster for its activity. A scaffold is defined as a protein that physically interacts with the desulfurase and cannot accept a Fe-S cluster from another protein (Py and Barras, 2010 Nature Review Microbiology). ISCA proteins: (i) can receive Fe-S from ISCU but cannot give their Fe-S to ISCU whereas they are able to transfer their cluster to different types of targets. Our experiments demonstrate that ISCA proteins behave as Fe-S carriers and that they can receive their Fe-S from the scaffold protein, ISCU. We have corrected the text accordingly.

However, the interpretation made of the phenotypes and identification of interacting partners is a more serious concern. In the interpretation of their work the authors state, or at least strongly imply, that the work of Shetel et al is flawed. The work of Shetel et al reported on the fact that a strong phenotype can be recognized in HeLa cells when either ISCA1 or ISCA2 is depleted and that the two proteins have the capacity to form a complex in vivo. Complex formation was also shown in the present work, also using a HeLa-based experimental system, but when using an entirely different experimental system the authors of the present work recognize a phenotype when ISCA1 is depleted but not when ISCA2 is depleted. From this they make the conclusion that "our results strongly suggest that HeLa cells or dividing cells may not be reliable to investigate the role of the proteins in physiological conditions and to decipher the molecular mechanisms driving the late mitochondrial Fe-S biogenesis". Exactly the reverse conclusion could also be made, after all Shetel et al do report a phenotype and it seems that a system for which a phenotype can be tracked would be a superior experimental tool. These comments are not intended to be hostile. Instead this reviewer, who is not a scientific competitor, believes that the overall tone of the manuscript, in its present form, does a disservice to the authors.

Response:

It was never our intention to suggest or strongly imply that the work of Sheftel et al is flawed, and we are grateful to the reviewer for pointing out that the tone of the manuscript gave this sense. The work of Sheftel et al provides evidence that in HeLa cells, ISCA1 and ISCA2 knockdown after multiple rounds of siRNA transfection leads to acidification of the cell culture media, gross alteration in intracellular morphology with large vacuolar structures, ultrastructurally abnormal mitochondria, a deficit of all mitochondrial Fe₄-S₄ cluster proteins tested and a deficit in a non Fe-S cluster mitochondrial protein COX2. These results indeed contrast with the results presented; however, the experimental settings are extremely different, HeLa cells are dividing cancer cells while skeletal muscle or neurons are post-mitotic cells and the morphological features observed in the HeLa cells strongly suggest cell suffering, due to strong mitochondrial deficiency, which most likely contributes to the phenotype (as pointed out by Sheftel et al). Re-reading through our manuscript with the reviewers comment in mind, we could see how our discussion could be misinterpreted. We have carefully revisited our

discussion to take into account the reviewers comments, and to change the tone of the discussion.

In the abstract the authors state "collectively, our data question the current view of late Fe-S mitochondrial machinery" although what, exactly is being questioned is not clear. There are indeed some very nice aspects that are reported and highly merit publication in a premier journal. For example, there is strong evidence that ISCA1 and ISCA2 are likely to have some different interacting partners. However, as confirmed in the work ISCA1 and ISCA2 also interact with each other *in vivo*. This strongly suggests a dynamic network, together with nice biochemical work, suggesting that ISCA forms are likely to harbor Fe-S clusters (although only shown in a heterologous system) and that ISCA proteins could be (not proven) acting as intermediate Fe-S carriers. It is unfortunate that, rather than concentrating on the novel observations, which underscore the complexity of the process, the authors choose to concentrate on emphasizing differences of their work with the published literature with an implicit suggestion that other approaches and suggestions are inherently flawed.

Response:

We have taken into consideration the comments of the reviewer and have revised the text accordingly. For example, we have changed the last sentence of the abstract to "Collectively, our data point to different requirements of ISCA1 and ISCA2 *in vivo*.", and the last sentence of the manuscript "Collectively, our results with the results in the literature strongly suggest that late mitochondrial Fe-S biogenesis is a complex and dynamic system that may have tissue and temporal specificity."

The title of the manuscript is misleading. It is not proven that "ISCA1 governs mitochondrial 4Fe-4S cluster biogenesis" as claimed. Rather it shows that mitochondrial 4Fe-4S cluster biogenesis can occur in the absence of ISCA2. It would be more appropriate to indicate that ISCA1 is essential for 4Fe-4S cluster synthesis and ISCA2 is not. After all, it could well be that the governing species is an ISCA1/ISCA2 hybrid and in the absence of ISCA2 its function can be replaced, in part or whole, by an ISCA1 X 2 species.

Response:

We have changed the title of the manuscript to take into consideration the comment of the reviewer. "ISCA1 is essential for mitochondrial Fe₄S₄ biogenesis while ISCA2 is dispensable under defined physiological conditions"

Also, the claim about "normal physiological conditions" would appear to be inappropriate given that the authors are using transfected cells that have shut down one or the other ISCA species Can this really be considered normal physiological conditions.

Response:

The physiological conditions, which could also be called *in vivo*, refers to experimental conditions within the living organism, in contrast to *in vitro* or *ex vivo* which are performed outside the living organism with artificial environment. All experiments performed in the mouse are under standard physiological condition, i.e. no stress. The physiological condition does not relate to the state of the cells when depleted. We have change "normal" to "standard" to clarify this point.

If ISCA is strictly a 2Fe-2S cluster carrier protein how is it that when it is produced heterologously in *E. coli*, where the native primary biosynthetic machinery is clearly not present it contains a 2Fe-2S cluster. It must be assumed in this situation that cluster-loaded IscU from *Escherichia coli* is the source of the cluster but this possibility is not explored in the present work.

Response:

We are aware that as we are in a heterologous expression system, the native biosynthetic machinery is not present. However, due to the highly conserved process across species, we speculate that the heterologously expressed mammalian ISCA proteins obtain the Fe-S cluster from the bacterial IscU. It would indeed be very interesting to understand how ISCA proteins obtain their cluster *in vivo*, and whether the cluster is provided by ISCU directly or via GLRX5 protein as suggested by *in vitro* experiments. However, this is beyond the scope of the present manuscript, but should be addressed in future experiments.

"Evolutionary" should probably be "Evolutionarily". Line 47 "consisting in" should probably be "consisting of".

Response:

Corrected

Reviewer #3 (Remarks to the Author):

Overview: Overall, this is a significant manuscript that may eventually be acceptable for publication. The authors have made significant advances in understanding the function of the IscA type protein that are associated with Fe/S cluster assembly. Mammals contain two IscA homologs, Isca1 and Isca2. Prior to this paper, the two proteins were viewed as being redundant, but the authors show that only Isca1 plays the role in Fe/S cluster assembly. They also show that both proteins bind Fe₂S₂ clusters and can transfer clusters. The experiments are generally presented well and the conclusions are reasonable. That being said, I do have some critical comments that must be addressed before I could recommend this paper for publication. I felt that the spectroscopy portion of the paper needs to be bolstered.

Comment 1: In Figure 1, whole cell spectra in C of the overexpression mutants show a small feature which is interpreted as the high-energy line of a quadrupole doublet due to [2Fe-2S]²⁺ clusters. However, the spectral intensity of this doublet represents only 10 - 12% of the total; most spectral intensity is due to high-spin Fe(II) and unresolved features in the middle of the spectrum that are attributed to Fe(III) nanoparticles. A lot of interpretive "weight" is being placed on a minor spectral feature, which is uncomfortable. How many replicate spectra have been obtained, and how reproducible is this minor spectral feature? If the experiment has only done once, or if the presented spectra are the best that they have collected, this must be repeated. If they have done the experiment multiple times, and see the same features, this should be mentioned in the text as it would strengthen their argument.

Comment 2: A related concern for the whole-cell spectra of Figure 1C is that the control spectra are significantly different from each other, with one showing substantial nanoparticles and the other showing less. Why? Again this is a bit unsettling. How many replicate spectra were obtained, and are these differences reproducible? The state at which the cells were harvested (log or post-log) could make a difference in nanoparticle formation.

Comment 3: In Table 1, the % of $[\text{Fe}_2\text{S}_2]^{2+}$ cluster in the control whole-cell samples is stated to be "0", but it must be possible to include some small percentage in a spectral simulation without a noticeable reduction in fit quality (my guess is that about 5% would fit in this way). Thus, it would be more accurate to state in the table something like "< 5%" rather than "0". But in this case, there is not much difference between control and overexpression spectra (e.g. 4% vs 11%).

Response:

These three remarks deal with the recordings and analyses of the *in cellulo* Mössbauer spectra, we therefore decided to answer the three questions together.

First of all, it must be underlined that no mutants of ISCA were overexpressed in cells. Figure 1C compares Mössbauer spectra recorded when ISCA1 (or ISCA2) is overexpressed (lower part) with those obtained without overexpression (top part).

We perfectly agree with reviewer #3 that a small contribution of oxidized $[\text{Fe}_2\text{S}_2]^{2+}$ cluster in the Mössbauer spectra of the control cell samples cannot be excluded. However, a contribution larger than 4% for such a cluster (versus the total iron content) strongly affects the theoretical spectra that no longer reproduce satisfyingly the experimental data. As a consequence, Table 1 was modified and a contribution less than 4% is now indicated for the $\text{Fe}_2\text{-S}_2$ cluster in control cell samples. It must be underlined that this contribution involves the TWO iron sites of the cluster (italic in Table 1), so that of EACH iron site is less than 2% (regular in Table 1, see caption of Table 1).

In order to present Mössbauer data in this manuscript, many samples including both control samples and those with overexpressed ISCA1 or ISCA2 have been prepared and Mössbauer spectra recorded. We indeed noticed some variations between the control cell sample spectra and simulations revealed a HS Fe^{II} contribution lying between 40 and 60%. Since a higher amount of HS Fe^{II} is observed when cells are put under anaerobic conditions (see for example *Biochemistry*, 2009, 48, 9234), this variation may be related to the action of dioxygen, during bacterial growth or centrifugation processes, despite our efforts to reproduce preparations under identical conditions. Formation of $[\text{Fe}_2\text{S}_2]^{2+}$ was reproducible and observed on overexpressed soluble ISCA samples. As indicated in Table 1, a contribution of 10 and 12% for EACH ferric site of the $\text{Fe}_2\text{-S}_2$ cluster is determined for ISCA2 and ISCA1, respectively. Accordingly, the TOTAL contributions of the Fe-S cluster amounts respectively to 20 and 24% versus the total iron content. Reviewer #3 must have been confused with the contribution of the individual site and that of the whole cluster. As a consequence, the $[\text{Fe}_2\text{S}_2]^{2+}$ content varies from less than 4% in control cell samples to at least 20% in overexpressed ISCA cell samples. This increase (16–20%) is significant and relevant.

In order to answer to reviewer 3, we have explored the possibility of increasing the spectral intensity signal by changing the expression level and condition of expression of the ISCA1 and ISCA2 proteins. We have successfully achieved this for ISCA2, as the protein is soluble and the expression is very reliable – with this new set of experiments, it brings the number of replicates for ISCA2 to 6 (4 replicates in the pET expression system that allowed us to detect 10-20% of $\text{Fe}_2\text{-S}_2$ (as presented in the initial submission) and 2 replicates using the pACYC expression system that allowed us to detect 28-30% of $\text{Fe}_2\text{-S}_2$ (new supplemental figure 1C and Table S1). All replicates confirm the initial report. Using the same strategy for ISCA1, it turned out to be more difficult. ISCA1 is a very unstable protein and tends to be insoluble when overexpressed. We were unable to obtain sufficient soluble ISCA1 protein using the pACYC expression system, we therefore turned back to the pET expression system in order to obtain an additional replicate.

We now have 3 replicates for the ISCA1 sample. All three experiments show Fe₂-S₂, above the limit of detection (between 8-24% of total iron), and the one presented in the original manuscript is the one with the best detection. We have added all this information in the revised manuscript to make it clearer to the reader.

Comment 4: I believe that the Figure 1 legend should say "applied parallel to the γ beam"

Response:

For Figure 1A, legend says applied perpendicular to the gamma-ray; for Figure 1C legend says applied parallel to the gamma-ray.

Comment 5: The authors report that they were unable to reduce the [Fe₂S₂]²⁺ clusters more than 15% - 20%. Hmmm. Have they tried this more than once? Have they tried dithionite at high pH? (I don't think the methods section states the pH of the dithionite solution. Dithionite is a more powerful reductant at high pH like 8.5). Did they examine the Mossbauer spectrum of the sample after treatment with dithionite? Was the remaining 80% - 85% of the cluster still intact and in the [Fe₂S₂]²⁺ state (or was there degradation? Or formation of magnetic material that might indicate a higher spin state??).

Response:

The reduction experiments were performed three times for each protein and we never obtained more than 15-20% reduction. When the experiment was performed using dithionite as reducing agent, a 0.1M stock solution of dithionite in 0.5 M Tris-HCl buffer at pH 8 was prepared (the final concentration used was 1 mM). The authors are aware that dithionite is a strong reducing agent at high pH and always use such conditions to reduce Fe-S proteins that are investigated in their laboratory. We agree with the referee that we have to better describe how the reduction of ISCA was performed (pH, ...) in the experimental section. This is done in the revised version.

We did not analyze by Mössbauer the samples after dithionite treatment. By EPR, no high spin state(s)(visible at low field) were visible. We tried to perform electrochemistry experiments to determine a redox potential but ISCA proteins were not stable and aggregated on the electrode. The aim of this EPR experiment was rather to confirm that ISCA proteins are Fe₂-S₂ proteins, which was the case. The authors would like to emphasize that it is the first time that an EPR signal is obtained on purified Fe-S ISCA proteins.

Comment 6: On page 6, line 124, I found it confusing that they mentioned that 78 K Mossbauer spectrum was "consistent with FeIII (S = 5/2) in tetrahedral environment..." but not to mention that it was also consistent with [Fe₂S₂]²⁺ clusters, which ended up supported by the 4.2 K data. They say that the parameters "unambiguously demonstrate" the presence of a [Fe₂S₂]²⁺ cluster with S = 0, but that seems too strong without any high-field data and simulation fit assuming diamagnetism.

Response:

We agree with the reviewer that the 78 K spectrum does not add much to the story and makes it more confusing. For these reasons, we decide to remove the 78K spectrum from the figure. The useful and required information is contained within the 4.2 K spectra.

The spectra were collected at 4.2 K in the presence of an external magnetic field of ca. 1.0 kG applied perpendicular to the gamma-rays. Indeed, this magnetic field is low. Nevertheless, these conditions are sufficient in the present case in order to safely identify the iron sulfur clusters. Let us first notice that the spectra contain

pretty narrow doublets and do not contain (within the noise level) additional iron sites. Therefore, the samples are fairly homogeneous.

1. The isomer shift and quadruple splitting values are consistent with Fe^{III} (S=5/2) ions in tetrahedral environment comprising sulfur ligands. Therefore, we exclude [Fe₄S₄]²⁺ and [Fe₃S₄]⁰⁺, [Fe₂S₂]¹⁺, [Fe₄S₄]¹⁺ clusters. Moreover, we exclude other ferric forms such as iron oxides.

2. The fact that we observe a quadruple doublet at 4.2 K in the presence of a small magnetic field indicates that these ions are involved in an exchange coupled system with a S = 0 ground state. Isolated Fe^{III} (S=5/2) ions or [Fe₃S₄]¹⁺ clusters are strictly excluded. Such systems give rise to magnetically split spectra in the presence of a magnetic field such as used here. For instance, formation of reduced [Fe₂S₂]¹⁺ (S=1/2) in overexpressed IscR cells was evidenced by features on the Mössbauer spectrum expanding between -4 and +4 mm/s (see *Biochemistry*, 51, 4453 (2012)). Note also that some EPR has been done before reducing with dithionite and the samples are EPR silent. This constitutes further evidence that the iron sulfur cluster is diamagnetic.

On the other hand both ISCA1 and ISCA2 are fully consistent with [Fe₂S₂]²⁺ clusters. The same approach with similar spectra was used in the paper by Mapolelo et al. (*Biochemistry*, 51, 8071 (2012)) to identify the [Fe₂S₂]²⁺ (S=0) cluster.

In addition, we have recorded Mössbauer spectra at 4.2 K with an external magnetic field of 7 T. The spectrum recorded on the overexpressed ISCA2 cell sample shows an additional contribution of a diamagnetic species, compared to the spectrum of the control cell sample. This signal is perfectly reproduced with the parameters of the [Fe₂S₂]²⁺ cluster listed in Table 1.

Comment 7: On page 7, the authors note that "in bacterial cells that overexpress Fe-S proteins, the iron content under the form of Fe-S never exceeded 20-40% of total iron." Is this a general "rule" or only the result found in the two cited reports? I've not heard of this before.

Response:

Sentence has been modified with respect to reviewer 3 comment.

Comment 8: The spectra in Figure 2 are somewhat confusing. In the text, they discuss mixing Holo-A1 and holo-A2 with apo-ISCU, and I expected that the corresponding spectra would be on the left (however they are on the right). More importantly, the text says that there was no cluster transfer but the data seem to show a 2-fold drop in the absorbance due to holo-ISCA2 during the experiment, and an increase in the absorption due to ISCU. Doesn't this suggest that a portion of the clusters were indeed transferred?

Response:

Figure 2A concerns only ISCA2 and ISCU. In view of the reviewers comment, we have changed the order of the ISCA2 presentation. On the left (upper and lower panels) are presented data when FeS-ISCA2 mixed with apo-ISCU whereas on the right (upper and lower panels) are presented data for apo-ISCA2 is mixed with FeS-ISCU. Transfer experiment with ISCA1 is presented in supplementary Figure 1C. The title of Figure 2 "Fe-S transfer experiments using ISCA1 and ISCA2 recombinant proteins" is a general title that includes (A)-(C).

Comment 9: Perhaps I missed it but I was unable to find an explanation of how the apo-forms of these proteins were prepared.

Response:

This is now included in the new version of the experimental part of the manuscript.

Comment 10: Have the authors monitored reduction of the clusters by UV-vis spectroscopy, using dithionite as the reductant? Can they determine whether the redox potential of the cluster is too low or whether reduction is kinetically sluggish? Why are they seeing partial reduction?

Response:

We have monitored reduction of both ISCA by UV-visible (see figures below with ISCA2 as example).

Spectra for both ISCA1 and ISCA2 are changed upon reduction with a loss of the absorption band in the 420-460 nm region. We observed after reduction with dithionite a characteristic shoulder at 550 nm, like for reduced ferredoxin, that is in agreement with a $[\text{Fe}_2\text{S}_2]^{1+}$ species.

For both proteins, no high spin state(s) at low field were detectable.

We have performed electrochemistry experiments to determine the redox potential of ISCA proteins but proteins were not stable and aggregated on the electrode.

We have tried to increase reduction time. Using dithionite longer incubation destroys the cluster (lower reduction yield and loss of iron after desalting column). 10 minutes was the best compromise we found to observe an EPR signal. Using DTT, the reduction is less efficient (DTT is not as strong as dithionite), and we have been able to observe a “nice” signal after 40 minutes which integrates for 15% of total iron. Therefore, the referee is probably right in the fact that there are likely some kinetics effects but we were able to observe EPR signal only using dithionite for around 10 min and DTT for 40 min. Again the authors would like to emphasize that the goal of these EPR experiments was to confirm that ISCA are $\text{Fe}_2\text{-S}_2$ proteins as observed by Mössbauer spectroscopy. From our experiments 15% corresponds to $\text{Fe}_2\text{-S}_2$ cluster in the +1 oxidation state. The rest is not known.

ISCA2 with dithionite (10 min): dashed line = after dithionite (1 mM, 10 min) reduction

ISCA2 + DTT (40 min): blue after DTT (50 mM, 40 min) reduction

Comment 11: In Figure 4, they seem to have repeated the Western analysis for a given column, but the figure legend doesn't mention this. Please clarify.

Response:

Replicates information has been included both in the legends and in methods section.

Comment 12: On page 13, the authors state "...but could shield a Fe₂S₂..." I was a bit confused regarding the word "shield". (bind??)

Response:

We have change the term shield for bind

Comment 13: There was a confusing shift in the paper and the authors need to reconcile and clarify this. In the first part of the paper, Isca1 and Isca2 proteins were both characterized and it seems that the lesson taught is that both are "viable" as Fe-S cluster carrier proteins, accepting a cluster from Iscu and donating it to a Fe₂S₂ ferredoxin. In fact, the evidence for Isca2 functioning in this regard is stronger than for Isca1. Both Fe/S loaded proteins could activate apo-aconitase implying the assembly of an Fe₄S₄ cluster. However, when the function of these proteins are investigated in mice and cell culture, Isca1 was found to be involved in mitochondrial Fe/S cluster assembly while the role of Isca2 is undefined but not involved in these processes. This apparent incongruency needs to be resolved or at least better clarified and explained by the authors. Left as is, this issue will cause confusion and it detracts from the impact of the paper.

Response:

We have to clarify this point. In vitro experiments show that ISCA1/ISCA2 contain Fe₂-S₂ and that they are Fe/S carrier proteins since they can transfer their clusters to both Fe₂-S₂ and Fe₄-S₄ proteins. Transfer to Fe₄-S₄ proteins is possible because DTT, a reducing agent, can reductively couple the two Fe₂-S₂ to form a 4Fe-4S. In vivo experiments performed on the present paper show that ISCA1 is essential for the maturation of Fe₄-S₄ protein whereas ISCA2 is not essential for that function under standard physiological conditions in the tissues tested. This is not in contradiction with the fact that ISCA2 transfer to Aco2 *in vitro*, experimental conditions devoid of any environmental parameters. It is possible that ISCA2 is involved in the maturation of Fe₄-S₄ proteins *in vivo* under specialized conditions that have to be defined. This was clarified in the revised manuscript.

Reviewer #1 (Remarks to the Author)

The authors have responded extensively to multiple reviewers. However, their responses do not allay concerns. For instance, there is no reference to Al-Hassnan, Z.N., Al-Dosary, M., Alfadhel, M., Faqeih, E.A., Alsagob, M., Kenana, R., Almass, R., Al-Harazi, O.S., Al-Hindi, H., Malibari, O.I., Almutari, F.B., Tulbah, S., Alhadeq, F., Al-Sheddi, T., Alamro, R., AlAsmari, A., Almunashri, M., Alshaalan, H., Al-Mohanna, F.A., Colak, D., and Kaya, N. (2015). ISCA2 mutation causes infantile neurodegenerative mitochondrial disorder. *J Med Genet* 52, 186-194, a paper in which a phenotype for ISCA2 mutations clearly appeared in patients. The paper is inconclusive in a field that has been marred by multiple misconceptions

Reviewer #2 (Remarks to the Author)

The reviewer appreciates the thorough and reasonable responses to the rather detailed comments of the reviewers. In the opinion of this reviewer the manuscript is significantly improved. There remains one stylistic comment for the authors consideration. There is a great deal of redundancy that appears in the introduction, results, and discussion sections. Much of this is not necessary and the manuscript could be streamlined, and improved, by eliminating some of the overlap.

Reviewer #3 (Remarks to the Author)

I found the authors' responses to be acceptable and recommend that the paper be accepted for publication in its current form.

Answers to Reviewers' comments:

Reviewer #1 (Remarks to the Author):

The authors have responded extensively to multiple reviewers. However, their responses do not allay concerns. For instance, there is no reference to Al-Hassnan, Z.N., Al-Dosary, M., Alfadhel, M., Faqeih, E.A., Alsagob, M., Kenana, R., Almass, R., Al-Harazi, O.S., Al-Hindi, H., Malibari, O.I., Almutari, F.B., Tulbah, S., Alhadeq, F., Al-Sheddi, T., Alamro, R., AlAsmari, A., Almontashri, M., Alshaalan, H., Al-Mohanna, F.A., Colak, D., and Kaya, N. (2015). ISCA2 mutation causes infantile neurodegenerative mitochondrial disorder. *J Med Genet* 52, 186-194, a paper in which a phenotype for ISCA2 mutations clearly appeared in patients. The paper is inconclusive in a field that has been marred by multiple misconceptions

Response: We regret that the reviewer feels that our responses do not alleviate his/her concerns regarding the manuscript. In response to the specific concern on the missing reference, this reference (corresponding to reference 44 – is now reference 47 in the revised version) was cited twice within the discussion, in both the initial and revised versions of the manuscript. The reference was used to discuss the point that in human, ISCA2 is essential and most likely required during development since a severe infantile mitochondrial disease is associated with mutations in ISCA2. Since reviewer 1 seems to have missed this reference and the corresponding discussion, we have modified the discussion and have combined the different aspects of the discussion about ISCA2 in a single paragraph at the end of the main text. We think these modifications clarify our view/discussion and better raise the questions that still need to be addressed to understand ISCA2 function and the complex process of Fe-S cluster biogenesis.

Reviewer #2 (Remarks to the Author):

The reviewer appreciates the thorough and reasonable responses to the rather detailed comments of the reviewers. In the opinion of this reviewer the manuscript is significantly improved. There remains one stylistic comment for the authors consideration. There is a great deal of redundancy that appears in the introduction, results, and discussion sections. Much of this is not necessary and the manuscript could be streamlined, and improved, by eliminating some of the overlap.

Response: We have taken the reviewer's comment into consideration and have revised the text accordingly. Specifically, we have reduced the last paragraph of the introduction, which in the original manuscript highlighted the results of the paper in great details. We have modified the first part of the results and combined the description of the biochemical characterization of ISCA1 and ISCA2 to avoid repetition of the technical description. We have made several modifications in the discussion to avoid repeating the results or minor points already discussed in the results. We believe that these modifications allow to streamline the manuscript by removing the redundancy. All the modifications are shown in the text using Word Track changes.

Reviewer #3 (Remarks to the Author):

I found the authors' responses to be acceptable and recommend that the paper be accepted for publication in its current form.

Reviewer #2 (Remarks to the Author)

This reviewer is satisfied with the minor changes in the manuscript as requested in the second review.